# Genome re-sequencing reveals the evolutionary history of peach fruit edibility

Yang Yu[1,2], Jun Fu[3], Yaoguang Xu[1,2], Jiewei Zhang[1,2], Fei Ren[1], Hongwei Zhao[3], Shilin Tian [4], Wei Guo[3], Xiaolong Tu[1,2], Jing Zhao[4], Dawei Jiang[4], Jianbo Zhao[1], Weiying Wu[1,2], Gaochao Wang[3], Rongcai Ma[1,2], Quan Jiang[1], Jianhua Wei[1,2] & Hua Xie [1,2]

Peach (*Prunus persica*) is an economically important fruit crop and a well-characterized model for studying *Prunus* species. Here we explore the evolutionary history of peach using a large-scale SNP data set generated from 58 high-coverage genomes of cultivated peach and closely related relatives, including 44 newly re-sequenced accessions and 14 accessions from a previous study. Our analyses suggest that peach originated about 2.47 Mya in southwest China in glacial refugia generated by the uplift of the Tibetan plateau. Our exploration of genomic selection signatures and demographic history supports the hypothesis that frugivore-mediated selection occurred several million years before the eventual human-mediated domestication of peach. We also identify a large set of SNPs and/or CNVs, and candidate genes associated with fruit texture, taste, size, and skin color, with implications for genomic-selection breeding in peach. Collectively, this study provides valuable information for understanding the evolution and domestication of perennial fruit tree crops.

[1] Beijing Academy of Agricultural and Forestry Sciences, Beijing 100097, China. [2] Beijing Key Laboratory of Agricultural Genetic Resources and Biotechnology, Beijing 100097, China. [3] Beijing 8omics Gene Technology Co. Ltd, Beijing 100083, China. [4] Novogene Bioinformatics Institute, Beijing 100015, China. These authors contributed equally: Yang Yu, Jun Fu. Correspondence and requests for materials should be addressed to H.X. (email: xiehua@baafs.net.cn) or to J.W. (email: weijianhua@baafs.net.cn) or to Q.J. (email: quanj@vip.sina.com)

The family Rosaceae includes many economically important perennial fruit tree crops of the genera *Prunus* (peach, almond, cherry, apricot, plum), *Malus* (apple), and *Pyrus* (pear). Recent genomic studies have advanced our knowledge of the evolution and domestication of many annual crops; however, such information is still limited for perennial fruit crops. Further, it is of great significance to elaborate the emergence and evolution of the fleshy and palatable fruits that are prized among the perennial fruit crops. Peach (*Prunus persica* (L.) Batsch, 2n = 2 × = 16, estimated genome size of 265 Mb) is a genetically well-characterized model for research about *Prunus* species and other Rosaceae fruit trees[1]. Domesticated peach in China was dispersed westward to Europe via the ancient Silk Road through Persia (present day Iran) in the final centuries B.C, and from Europe to the Americas during the 16th century[2] (Fig. 1a). Today, peach is widely cultivated in temperate and subtropical zones throughout the world, with an annual global production in excess of 20 million tonnes[3].

The direct wild ancestor of peach remains unknown, and is likely extinct, so there is controversy about the origin and evolutionary history of this fruit. It has been proposed that peach was originally domesticated in northwest China around 4000–5000 years ago[2], while, fossil evidence indicates that peach cultivation and domestication could date back to at least 7500 years ago in the Yangtze River valley of southern China[4]. Surprisingly, peach endocarp fossils from 2.6-million-year-ago (Mya), found recently in Kunming, southwest China, are indistinguishable from endocarps of the modern peach cultivars, and studying of these fossils suggested that peaches may have acquired their modern-like edible fruits long prior to domestication, perhaps mediated by frugivorous primates[5]. It is thus possible that there was a very long period of pre-selection that occurred in natural environments that may have enabled the development of edible fruit (fleshy and palatable mesocarp) long before outward dispersal for cultivation and domestication.

*Prunus* subgenus *Amygdalus* comprises two sections, *Amygdalus* and *Persica*, in which domesticated almond and peach are classified, respectively. The common ancestor of this subgenus likely bears a dry, splitting mesocarp, while the transition to fleshy and palatable mesocarp in some *Persica* section species (wild *P. mira* and *P. kansusensis*, and *P. persica*) (Supplementary Table 1) was presumably derived from selection and domestication in China[6,7]. Great morphological variation in fruit traits like fruit size, texture, taste, and skin color among cultivated peach and its wild relatives offers a natural diversity panel that presents an opportunity to explore the speciation and domestication history of peach, and to interpret the emergence and evolution of fruit edibility in perennial fruit crops more generally. Previous genome studies with many cultivated peaches but limited number of wild relative peaches have identified genomic regions that have undergone artificial selection, which have impacted the diversification of several phenotypes in peach[8,9]. However, as there is only limited genomic data available for investigating the ancient speciation or early domestication history of peach, little is known about the evolution of fruit traits of modern peaches.

Here, a large-scale SNP data set from 58 high-coverage genomes of cultivated peach and closely related relatives, including 44 newly re-sequenced accessions from this study and 14 accessions from a previous study[10], was used to explore the evolutionary history of peach. In addition to clarifying phylogenetic relationships of cultivated peach and its close relatives, our genomic analyses highlight that the diversification of several peach species is closely related to topographical changes caused by a series of uplifts of the Tibetan plateau. Our study reveals the genetic basis of the emergence and evolution of fruit edibility in peach (*Persica* section) species and supports the hypothesis that

frugivore-mediated selection may have occurred[5] several million years before the eventual domestication of peach. Further, to gain insight into peach domestication, we identified genomic regions with strong selective signatures related to fruit texture, taste, size, and skin color, and propose that selection for fruit size occurred prior to the selection for fruit skin color that occurred during peach modern improvement.

## Results

**Genetic divergence of peach and closely related relatives.** Previous studies support that peach was domesticated in China[2], while the emerged fleshy mesocarp phenotypes are found not only in *P. persica* but also in *P. mira* and *P. kansusensis*. These facts suggested a way to explore the selection and/or domestication history that has driven the transition to fleshy edible fruit in some *Persica* section species. To clarify the taxonomy of the subg. *Amygdalus* species endemic to China and to study the evolved fruit edibility in peach species, we generated a total of 700 Gb of raw sequence data for 44 high-coverage genomes of cultivated peach and all known wild relative species endemic to China, and its closely related species (Supplementary Table 2). Approximately 677 Gb of high quality reads (Supplementary Table 2) were aligned against the reference genome with a mapping rate ranging from 82.24 to 96.11%, resulting in coverage depths ranging from 41.25 to 72.18 × (Supplementary Fig. 1 and Supplementary Data 1). Combining our data with 14 additional peach genomes[10] (all with >10× coverage depth) from Verde et al.[10], we identified a total of 24,280,369 SNPs in the 58 accessions (Supplementary Data 2). The accuracy of called SNPs was assessed using a customized peach SNP array with 27,862 probes (Supplementary Data 3), and was estimated to range from 97.36 to 98.12% (Supplementary Table 3).

We conducted a neighbor-joining (NJ) phylogenetic analysis and a principal component analysis (PCA) of the 58 accessions and *P. mume* (subg. *Prunus*) (Supplementary Fig. 2). Both analyses clearly placed all cultivated peaches (CP) (*P. persica* and *P. ferganensis*) and wild relative peaches (WP) (*P. mira*, *P. davidiana*, *P. davidiana* var. *potanini*, *P. tangutica*, *P. mongolica*, *P. kansuensis*) into *Persica* section of subg. *Amygdalus*, placed only cultivated almonds (CA) (*P. dulcis*) into *Amygdalus* section of subg. *Amygdalus*, and placed the three other species (*P. ledebouriana*, *P. triloba*, and *P. pedunculata*) outside subg. *Amygdalus* (Supplementary Fig. 2a,b and Supplementary Note 1). Removing the non-*Amygdalus* accessions, we performed PCA (Fig. 1b) and generated a NJ tree for the 51 *Amygdalus* accessions (Fig. 2). The first two principal components PC1 and PC2 clearly differentiated the CA accessions from the CP and WP accessions, and differentiated the peaches into separate groupings of the accessions: CP, "WK" (*P. kansuensis*), "WM" (*P. mira*), and "WD" (*P. davidiana*, *P. davidiana* var. *potanini*, *P. tangutica*, and *P. mongolica*).

Median-joint network analysis (Supplementary Fig. 3), identity score, and identity-by-state analyses (Supplementary Fig. 4) each verified these divergence patterns. According to our analysis, almond is phylogenetically far from peach species, a finding that appears to counter Charles Darwin's speculation that peaches are perhaps almonds that have been modified in marvelous manner[11]. Our results also add robust support that *P. tangutica* and *P. mongolica* should be classified into the *Persica* section of subg. *Amygdalus*[6,7], although these two species bear split, thin, and dry mesocarps and are morphologically similar to *P. dulcis*. Thus, the WD grouping contains all of the inedible peach species in the *Persica* section.

In the NJ tree using *P. ledebouriana* as the outgroup, the WM accessions were first split from other peaches, followed by the WD accessions, and then the WK accessions in the *Persica*

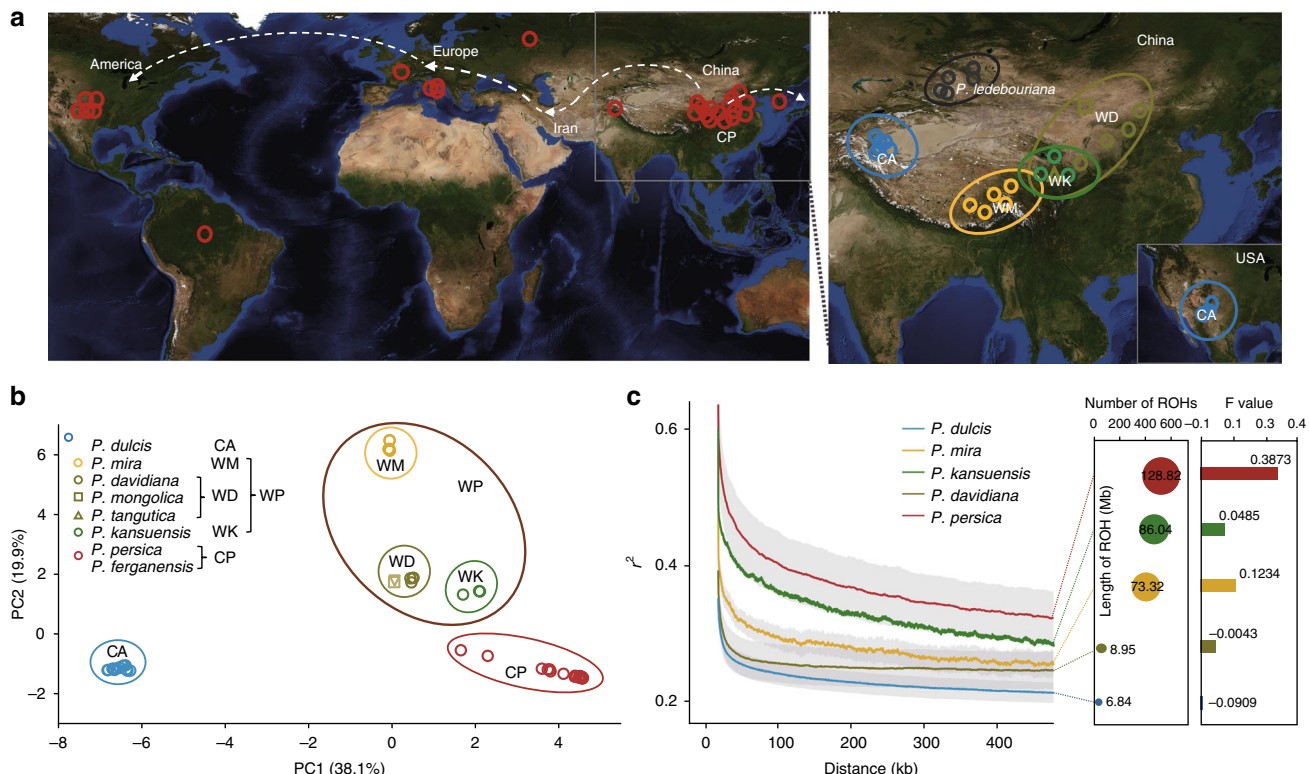

**Fig. 1** Genetic divergence of cultivated and wild relative peach, and cultivated almond. **a** Geographic distributions of the *Amygdalus* accessions altogether with *P. ledebouriana*. The CP group includes 22 *P. persica* and *P. ferganensis* accessions (left); the WP group includes 29 WM (*P. mira*), WD (*P. davidiana*, *P. davidiana* var. *potanini*, *P. tangutica*, and *P. mongolica*), and WK (*P. kansuensis*) accessions; the CA group includes 15 *P. dulcis* accessions, as well as 5 *P. ledebouriana* accessions (right). The locations of all the accessions are shown with colored symbols corresponding to Fig. 1b. The dispersal route of domesticated peach was shown with white dotted lines. The topographic map was obtained from NASA.gov. **b** PCA plots of the first two principal components using a total of 2,559,589 whole-genome SNPs (maf<5%, call rate>90%). The first two principal components PC1 and PC2 differentiated the CA accessions from the CP accessions and the WP accessions, and clustered the peaches into four groupings: CP, WK, WD, and WM. **c** Linkage disequilibrium patterns of five typical species: *P. persica*, *P. kansuensis*, *P. davidiana*, *P. mira*, and *P. dulcis* based on the SNPs from the accessions sequenced in this study and data downloaded from Verde et al.[10] and Cao et al.[8] (Supplementary Data 1). Patterns of ROHs (total number, and length indicated by circled areas) and *F* values of the five species are shown

section, and the CP accessions were further grouped into two subgroups designated as "PL" (*persica* landraces) and "PMC" (*persica* modern cultivars) (Fig. 2), supporting the conclusion that cultivated peach originated from a single domestication event[8]. We performed model-based clustering analysis using ADMIXTURE[12], and the same groupings were identified among the *Amygdalus* accessions ($K = 9$) (Supplementary Fig. 5a). Interestingly, we found a unique grouping containing all of the *P. ferganensis* accessions ($K = 9$) that were clustered with the *P. persica* landraces from north China ($K = 8$), supporting that the edible *P. ferganensis* is a geographical ecotype of *P. persica*[8]. ADMIXTURE, which is widely used for investigating the population structures of many domesticated crops assumes that loci are at HWE (Hardy–Weinberg equilibrium) within populations. Considering the potential impact of HWE violations, we also filtered SNPs by testing HWE violation ($P > 10^{-4}$) and performed model-based clustering analysis; this analysis gave to a consistent conclusion about the population structure ($K = 9$; Supplementary Fig. 5b). We determined the number, frequency per kb, and heterozygosity of SNPs for the CP, WP, and CA accessions (Fig. 2 and Supplementary Fig. 6), and found that fewer than ~5% of SNPs were common to all three groups (Supplementary Fig. 7); we also noted high $F_{ST}$ values among these three major groups (Supplementary Table 4), highlighting their extensive genetic differences.

**Genomic footprints of selection and domestication.** Given that the ancestor of *Amygdalus* and *Persica* species had a dry, splitting, and inedible mesocarp[6], it has been proposed that the transition into a non-splitting fleshy mesocarp—as found in *P. mira*, *P. kansuensis*, and *P. persica* of *Persica* section species—likely resulted from selection and/or domestication[6,7]. Notably, we observed an increase in the ratio of nonsynonymous to synonymous SNPs and noted the lowest MAF (MAF < 0.1) values for domesticated CP accessions as compared with the WP and CA accessions (Fig. 2, Supplementary Fig. 8 and Supplementary Data 2), indicating that positive selection and a population bottleneck occurred during the domestication of cultivated peach. The domestication caused a reduction of genetic diversity: the ratio ($\theta\pi_{WP}/\theta\pi_{PL} = 5.1485$) of WP to PL is within the range of genetic bottlenecks characterized by the domestication of other fruit and grain species ($\theta\pi_{W}/\theta\pi_{C} = 1.20–5.43$)[13], confirming that a severe bottleneck occurred during peach domestication. Thus, peach domestication appears to be distinct from the domestication of other perennial species like grapevine and apple, both of which experienced mild bottlenecks[14,15]. Although very little reduction of diversity ($\theta\pi_{PL}/\theta\pi_{PMC} = 1.0004$) occurred during peach improvement, the much lower Tajima' D value of PMC (0.0078) as compared with PL (0.5218) indicates that modern peaches have experienced strong artificial selection during peach improvement (Supplementary Table 4).

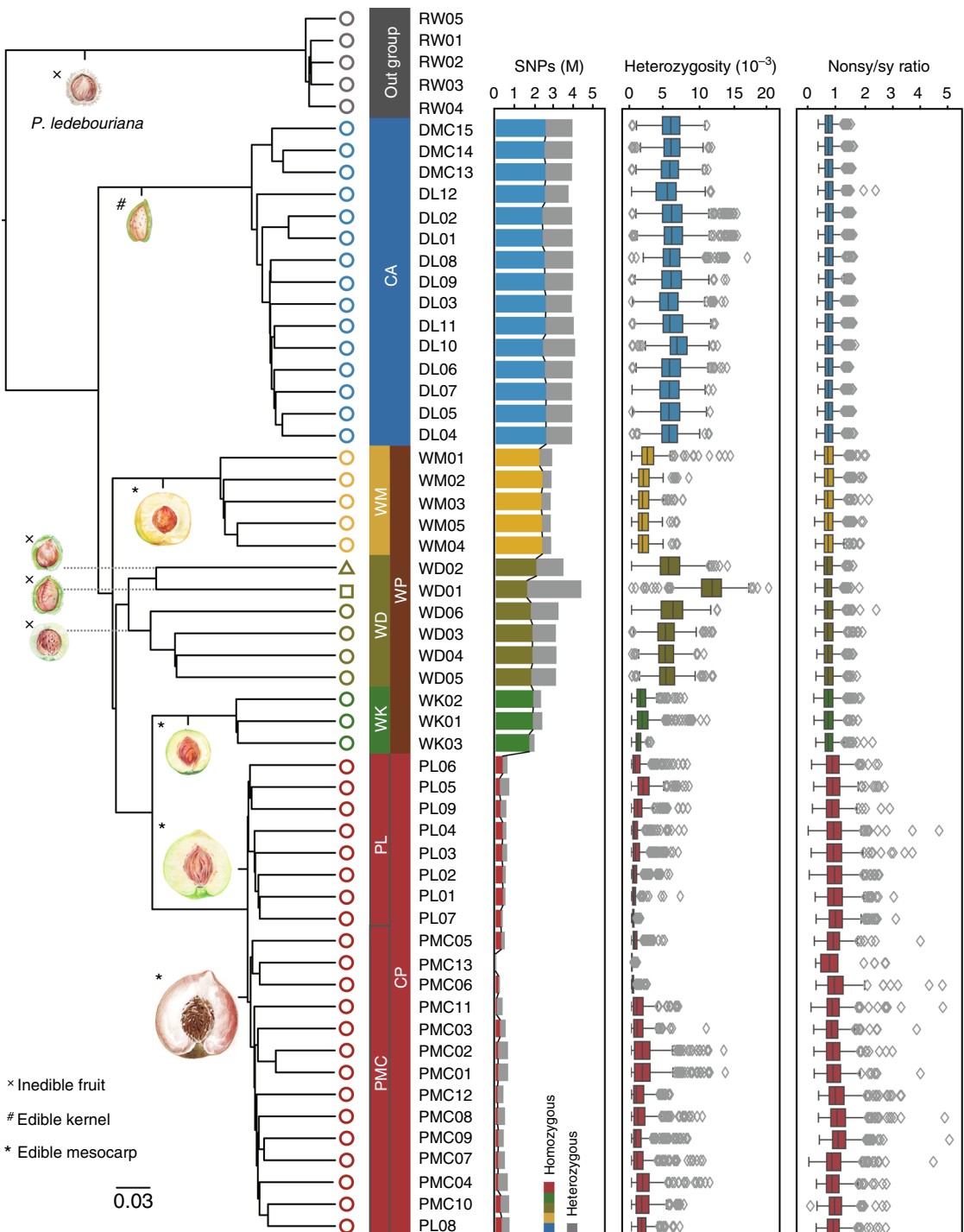

**Fig. 2** Neighbor-joining phylogenetic tree of the 51 *Amygdalus* accessions. The NJ tree altogether with the number, heterozygosity, and Nonsy/sy ratio of SNPs for all the *Amygdalus* accessions were shown. The NJ tree was generated using *P. ledebouriana* as the outgroup based on an analysis of a total of 3,909,617 whole-genome SNPs (maf<5%, call rate>90%). Horizontal lines at the center of each box show median values. Bounds of each box show the quartiles. The upper whisker extends to last datum less than Q3 + 1.5*IQR, where IQR is the interquartile range (Q3 - Q1). The lower whisker extends to last datum larger than Q3 - 1.5*IQR. The typical fruit morphology and edibility of all the *Amygdalus* species and the outgroup species in this study are shown. Fruit images were created by Yang Yu and Shan Gao

It is important to note that, compared to *P. davidiana*, and *P. dulcis*, there were increased inbreeding levels in the three edible-fruit peach species (*P. persica*, *P. kansuensis*, and *P. mira*). This trend was supported by three separate lines of evidence, including analyses of the total length of ROH (regions of homozygosity), inbreeding coefficients (*F* value), and heterozygosity (Fig. 1c,

Supplementary Table 5). So the increased inbreeding level and the transition to fleshy fruit represent two domestication syndromes that possibly resulted from selection and/or domestication during the evolution of peach species (Supplementary Note 2). Linkage disequilibrium (LD) is influenced by recombination rates, mutation rates, genetic drift, inbreeding, population structure,

genetic linkage, and selection; LD analysis based on SNP data from the genome sequences of the present study and a previous study[8] indicate that *P. persica* has a much slower decay of pairwise correlation coefficient ($r^2$) values than *P. kansuensis*, *P. davidiana*, *P. mira*, or *P. dulcis* (Fig. 1c). It has been reported that self-fertilization in peach promotes the maintenance of extended LD[16], so the increased extent of LD appears to have been impacted by the inbreeding in the evolutionary history of *Persica* section species.

**Divergence events and demographic history**. To explore the evolutionary history of peach, we used Bayesian phylogenetic analysis of molecular sequences (BEAST) analysis with a

substitution rate of $7.7 \times 10^{-9}$ per site per generation[17] to investigate the divergence times of *P. persica*, *P. kansuensis*, *P. davidiana*, *P. mira*, and *P. dulcis* (Fig. 3). According to our estimation, peaches and almonds diverged from a common ancestor about 4.99 Mya (95% confidence intervals (CI): 4.79–5.19), a result similar to a previous estimation of 5.09 Mya[18]. The Central Asian Massif-uplift occurred during this period, and may have been the vicariance event that gave rise to the observed geo-graphical distributions and contributed to the distinct phenotypic features of *Amygdalus* section species and *Persica* section species[7].

The subsequent peach divergence patterns for *P. mira*, *P. davidiana*, *P. kansuensis*, and *P. persica* species appear to

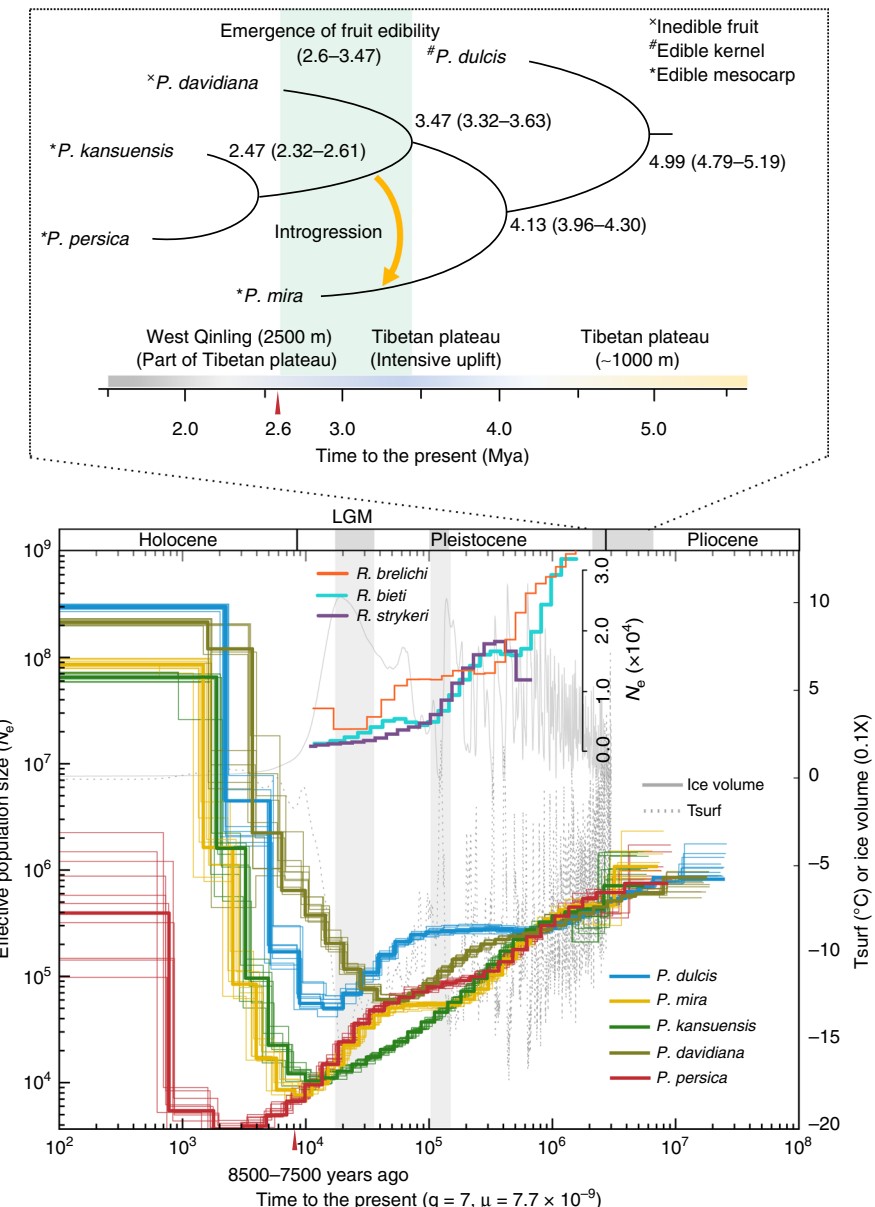

**Fig. 3** Speciation and demographic history of peach species. BEAST analysis was used to estimate divergence times of *P. persica*, *P. kansuensis*, *P. davidiana*, *P. mira*, and *P. dulcis*, and an MSMC model was used to infer their effective population ($N_e$) fluctuations under a mutation rate $\mu = 7.7 \times 10^{-9}$ per site per generation according to Xie et al.[17]. The divergence time of each species was shown in each branch, the emergence time of edible fruit in peach species (4.13–2.6 Mya) was showed with green shadow, and a single introgression event is indicated with an orange arrow between *P. mira* and the common ancestor of *P. kansuensis* and *P. persica*. Peach speciation likely was influenced by uplifts of the Tibet plateau. In the demographic diagram, the gray solid and dotted lines show, respectively, the ice volume and the global surface air temperature (Tsurf), as reconstructed using benthic oxygen isotope (δO[18]) stack record data[70], and the $N_e$ trajectories of three monkey species (*Rhinopithecus. brelichi*, *R. bieti*, and *R. strykeri*) from Zhou et al.[28] are inserted in the diagram. Two red triangles indicate peach endocarp fossil evidence from 2.6 Mya and 8500–7500 years ago, respectively[4,5]. LGM, Last Glacial Maximum

have been strongly influenced by serial uplift events of the Tibetan plateau. These uplifts are known to have resulted in a distinct geographic and climatic transition[19] that led to the speciation of a large number of plant taxa in southwest China, where there is high species diversity for *Rosaceous* genera[5]. The emergence of *P. mira* (the oldest wild ancestor native to the Tibetan plateau) occurred 4.13 Mya (95% CI: 3.96–4.30) when the Tibetan plateau surface was characterized by gentle undulations at ~1000 m altitude[19]. *P. davidiana* emerged about 3.47 Mya (95% CI: 3.32–3.63) when a large-scale intensive uplift of the Tibetan plateau turned northwest China into an arid inland region;[20] this uplift likely led to the allopatric speciation of the drought-tolerant *P. davidiana*. *P. kansuensis*, which is native to a region that is geographically close to the West Qinling Mountains (a part of the northeast Tibetan plateau), diverged about 2.47 Mya (95% CI: 2.32–2.61) from the population that also gave rise to *P. persica*. This divergence was likely influenced by the uplift of the West Qinling Mountains to an altitude of 2500 m[21] and was likely fostered in this deciduous forest ecoregion that is known to have supported a huge variety of plants.

These results led us to infer that *P. persica* originated in southwest China, the subtropical forested region south of the West Qinling Mountains that had a warm-humid climate at that time. If this is the case, it could help explain the discovery of the peach endocarp fossils (from 2.6-million-year-ago) in Kunming, southwest China[5]. Intriguingly, these fossil endocarps are morphologically identical to the endocarps of modern *P. persica* descendants;[5] specifically, the fossil endocarps suggest a mean fruit diameter of ~5.2 cm, which is comparable to the size of modern peaches and is remarkably larger than the fruit of extant non-edible peach species. Based on an estimated time to the most recent common ancestor by estimating the divergence time between PL and PMC using all accessions in these two subgroups, we inferred that the unknown direct wild ancestor of cultivated *P. persica* originated at least 0.60 Mya in the Pleistocene. Together, these results indicate that the emergence time of edible fruit in peach species occurred between 4.13–2.6 Mya and the direct wild ancestor of cultivated *P. persica* emerged between 2.47–0.60 Mya.

We used multiple sequentially Markovian coalescent (MSMC) analysis to investigate the population histories of *P. dulcis*, *P. mira*, *P. davidiana*, *P. kansuensis*, and *P. persica*, and interpreted these results in light of known geological and climatological trends ever since Pliocene to Holocene (Fig. 3). Notably, starting from the last interglacial period (130–115 Kya), the effective population size ($N_e$) of *P. mira* and *P. persica* decreased less precipitously than the $N_e$ of either *P. kansuensis* or *P. davidiana* (Fig. 3). We speculate that the observed reduction in the rate of decline in *P. mira* and/or *P. persica* may resulted from relatively mild climate oscillations in their specific habitats in southwest China, which coincide with known glacial refugia that resulted from the uplifts of the Tibetan plateau in the late Pleistocene[19]. Thus, it is conceivable that the *P. mira* and *P. persica* populations were relatively better protected in the refugia than the *P. davidiana* and *P. kansuensis* populations outside these areas. By the time of the Last Glacial Maximum (26.5–19.0 Kya), a period of extremely low temperatures, both the *P. mira* and *P. persica* populations again experienced precipitous declines. Given the highly similar trajectories of *P. mira* and *P. persica* before the Holocene, we posit that *P. mira* and *P. persica* shared overlapping habitats in southwest China before the dispersal of *P. persica* to other regions. Our results support that southwest China provided an evolutionary cradle for peach and ensured the possibility of uninterrupted interactions (perhaps selection) between animals and plants.

Subsequently, the stable rewarming of the global climate that occurred during the Holocene would have expanded the area of suitable habitats for these five *Prunus* species. The $N_e$ of all these species increased during this period with the exception of *P. persica*, which had a delayed increase in $N_e$. The population expansion of *P. mira* and *P. kansuensis* began about 10,000 years ago, while *P. persica* was experiencing a continuous and severe reduction of $N_e$ that lasted until about 2000 years ago. This reduction was likely due to a founder-effect bottleneck that resulted from its outward dispersal from southwest China for cultivation. This scenario helps explain archeological findings which indicate that cultivated peach was present 8500–7500 years ago in the substantial human settlements of the Yangzi valleys of southern China[4]. The recent expansion of the *P. persica* population that began about 2000 years ago (Fig. 3) likely resulted from extensive agricultural management, a scenario that is consistent with the historical record found in the "Classic of Poetry" which suggests that peach cultivation was occurring *circa* 2500 years ago.

**Early emergence of peach fruit edibility**. Using the TreeMix[22] model we detected significant introgression from *P. kansuensis* to *P. mira* (Fig. 4a), and this introgression between *P. kansuensis* and *P. mira* was supported by a significant excess of shared derived alleles (Fig. 4b). We also found an excess of shared derived alleles (Fig. 4b) between *P. mira* and *P. persica*. The introgression between *P. mira* and *P. persica* was not significant at the genome-wide level, likely because introgressed segments were lost during domestication. Modified *D* tests identified 302 and 217 introgressed segments respectively, between *P. mira* and *P. kansuensis* and between *P. mira* and *P. persica* (Supplementary Fig. 9 and Supplementary Data 4). Significantly reduced sequence divergence ($d_{XY}$) for these introgressed segments was found as compared with the genome background (Fig. 4c), further supporting that these segments resulted from introgression rather than from shared ancestry from the common ancestor of *P. mira* and *P. dulcis*. Intriguingly, we found 320 predicted genes in the common introgressed segments; a number much higher than the expected average (26 genes) predicted based on random sampling (1000×) (Fig. 4d). Based on the genes from the common introgressed segments, both the introgression between *P. mira* and *P. persica* estimated at 3.58 Mya (95% CI: 3.35–3.83) and the introgression between *P. mira* and *P. kansuensis* estimated at 3.65 Mya (95% CI: 3.40–3.91) predate the divergence of *P. kansuensis* and *P. persica* (2.47 Mya), supporting the conclusion that the excess of common introgressed segments we observed result from an ancient introgression between *P. mira* and the common ancestor of *P. kansuensis* and *P. persica*.

Compared to the inedible *P. davidiana*, we found significantly stronger selection signatures in the introgressed segments than the genome average levels of *P. mira* (*P*-value < 5.92 e−22, *t*-test), *P. kansuensis* (*P*-value < 9.08 e−120, *t*-test), and *P. persica* (*P*-value < 6.14 e−24, *t*-test) using XP-EHH[23]. We analyzed significantly overrepresented gene ontology (GO) terms among the 997 and 703 predicted genes of the two introgressions, respectively (Supplementary Data 5). Among the enriched terms (*P*-value < 0.05, Fisher's exact test), we found various categories putatively associated with fruit fleshiness and palatability (Fig. 4d and Supplementary Data 6 and 7). Notably, we found endomembrane trafficking related genes[24] encoding the Sec23/Sec24 protein transport family proteins, vacuolar ATP synthase subunits, SCAMP family protein, and cell-wall modification related genes encoding pectin lyase-like protein, plant invertase/pectin methylesterase inhibitor, pectinesterase; all of these have been associated with fruit texture[24–26]. We also identified genes encoding sucrose-6F-phosphate phosphohydrolase, glyceraldehyde 3-phosphate dehydrogenase, and glycosyl hydrolase, which

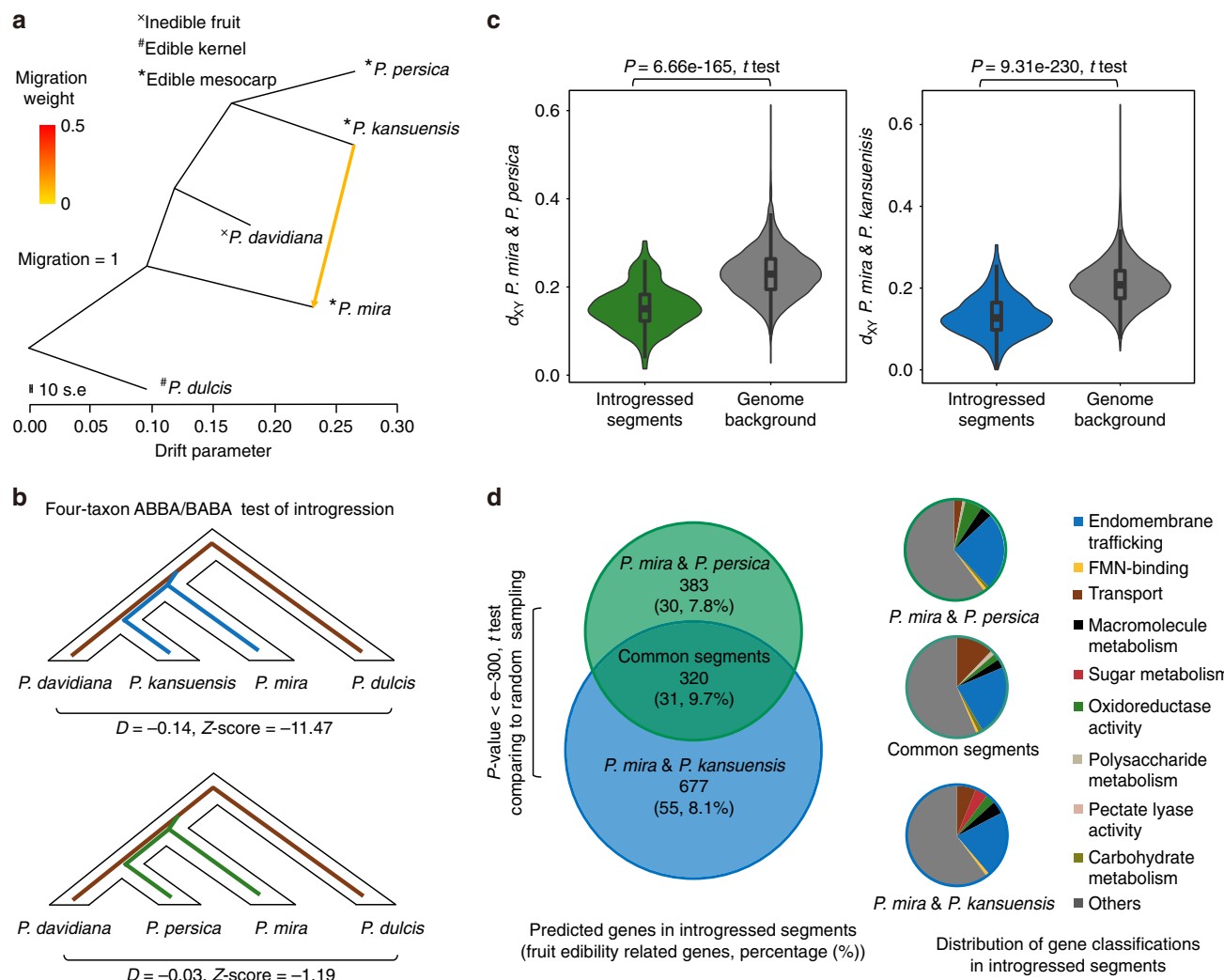

**Fig. 4** The emergence of fruit edibility in peach species. **a** TreeMix analysis of *P. mira*, *P. davidiana*, *P. kansuensis*, and *P. persica*, using *P. dulcis* as the outgroup. A significant gene flow from *P. kansuensis* to *P. mira* was observed when migration = 1. **b** Four-taxon ABBA/BABA test of introgression. The excess of shared derived alleles (indicated by *D* values and *Z*-scores) were observed between *P. kansuensis* and *P. persica*, and between *P. mira* and *P. persica*. **c** $d_{XY}$ values of the introgressed segments and genome background for introgression between *P. mira* and *P. kansuensis* and between *P. mira* and *P. persica*. Significantly lower $d_{XY}$ values were observed in introgressed segments compared to the genome background. Horizontal lines at the center of each box show median values. Bounds of each box show the quartiles. The upper whisker extends to last datum less than Q3 + 1.5*IQR, where IQR is the interquartile range (Q3 - Q1). The lower whisker extends to last datum larger than Q3 - 1.5*IQR. **d** Analyses of the predicted genes embedded in introgressed segments. The number and the proportion of putative genes related to fruit edibility within introgressed segments are given inside the parentheses of the Venn diagram. Functional categories containing significant enrichment of gene ontology (GO) terms were presented using pie charts. Significantly enriched (*P*-value < 0.05, Fisher's exact test) terms related to fruit edibility were classified into categories (Supplementary Data 6) based on their relatedness in the ontology or on functional proximity

have been associated with sugar metabolism[25,27]. These data suggested that fleshiness and palatability, which are the basic characteristics of fruit edibility, originated in the common ancestor of *P. kansuensis* and *P. persica* and were then transferred to *P. mira* before being subsequently inherited by *P. kansuensis* and *P. persica*. Therefore, the emergence of fruit edibility in peach species should date to 3.47–2.6 Mya (Fig. 3), rather than about 4.13 Mya (the time when *P. mira* diverged).

As we studied the evolution process of fruit edibility, we noted a study of three monkey species[28] that offers an interesting parallel to our demographic results (Fig. 3). The $N_e$ trajectories of the two species that inhabited in southwest China (*Rhinopithecus brelichi* and *R. bieti*) were very similar to those of *P. mira*, and *P. persica*, while the $N_e$ trajectory of the other species from a different location (*R. strykeri*) displayed no such synchronous

demographic changes[28]. We used a Granger causality (GC) test[29] to assess if there is a similar trend in the effective population size ($N_e$) between the three monkey species and the peach species. Intriguingly, our GC results implied that the $N_e$ of the two monkey species (*Rhinopithecus brelichi* and *R. bieti*) can be used to predict the $N_e$ trajectories of the three edible peach species (*P. mira*, *P. kansuensis*, and *P. persica*) (Supplementary Note 3). Beyond sharing similar trajectories of estimated population fluctuations in the same region, monkey species like *R. brelichi* and *R. bieti* may have used edible *P. mira*, *P. kansuensis*, and *P. persica* as food, perhaps exerting selection pressure on edible peaches; this is also supported by the fact that these three edible peach species to some extent exhibit a "selection footprint", both morphologically (thicker and fleshy mesocarp) and genetically (increased inbreeding level) (Supplementary Note 2). Therefore,

we propose that the ancient selection process likely involved frugivores and contributed to the emergence of fruit edibility prior to human-mediated domestication. Our study provides molecular evidence for a frugivore-mediated selection that contributed to the dispersal syndrome (suites of fruit and seed characteristics that attract dispersers)[30].

**Enhanced fruit texture and taste during peach domestication.** Fruit texture and taste are two major targets of domestication for most fruit crops. We searched for genome regions bearing strong selective signatures using a number of comparisons with the CP group and with the PL subgroup vs. the WP group and vs. a group that comprised accessions from WD and WK, the two most closely related groups to cultivated peach (Supplementary Data 8). We analyzed significantly ($P$-value < 0.05, Fisher's exact test) overrepresented GO terms among the predicted genes within selective sweep regions (Supplementary Data 9) and found enrichment of putative fruit edibility related terms associated with cell wall modification, sugar metabolism, and ion transport (Supplementary Data 10). In these enrichment terms, we identified genes encoding cell wall synthases, cell wall lyases, and pectin lyase-like proteins that are known to contribute to fruit texture;[31–33] genes known to function in the synthesis and transport of sugars that have been reported as sugar-metabolism related QTLs which have contributed to sweetness in peach;[34] and genes encoding ion transport proteins that have been implicated in mesocarp development[35] (Supplementary Fig. 10 and 11 and Supplementary Data 11 and 12).

Copy number variations (CNVs) are also known to be the targets of artificial selection. We characterized the CNVs in each of the accessions with high sequencing coverage (Supplementary Table 6), and adopted $V_{ST}$ to assess genome regions that putatively experienced artificial selection during peach domestication and/or improvement. We identified genes located within regions exhibiting high $V_{ST}$ values (top 5%) (Supplementary

Data 13 and 14) and performed GO enrichment analysis of these genes to characterize their functional annotations, and noted enriched GO terms ($P$-value < 0.05, Fisher's exact test) potentially associated with fruit edibility (e.g., "lyase activity" (GO:0016829), "hydrolase activity" (GO:0016787)) (Supplementary Data 15 and 16).

Of more direct interest, we found these selective regions harbored genes likely related to fruit texture, including genes encoding expansin proteins, pectin lyase-like proteins, cinnamyl alcohol dehydrogenases, and a polygalacturonase[31,36–38], and genes that likely contribute to fruit taste, including genes encoding a polyol/monosaccharide transporter[39] (Supplementary Figs. 12–13 and Supplementary Data 16). Notably, we found that orthologs of *EXPA16*, *Pectin lyase-like*, *CAD9*, and *PMT5* increased in copy number in *P. persica* in the course of domestication and/or subsequent improvement (Fig. 5), suggesting that a gene dosage effect may have contributed to the fruit texture and taste of modern peaches. Studies in both plants and animals have demonstrated that CNVs have had a particularly large influence on quantitative traits[40,41]. Thus, our results indicate that CNVs have impacted quantitative traits for fruit edibility (e.g., fruit texture and taste) during peach domestication and subsequent improvement.

**Stage-wise selection for peach fruit size and skin color.** Increased fruit size is considered to be an evident domestication syndrome of fruit crop domestication. A conspicuous increase in fruit size occurred during human-mediated domestication of *P. persica*, and given that peach stone size is strongly correlated with peach fruit size[4], fossil evidence of increasing peach stone sizes from 8000 to 3500 years ago also provides a "visualized" domestication process of fruit size in peach[4]. The *FW2.2/CNR* family genes are known to affect the fruit size in multiple fruit species[42–44]. Using selective sweep analysis, we identified two *CNR* genes (*PpCNR13*, *PpCNR17*) (Supplementary Fig. 10 and

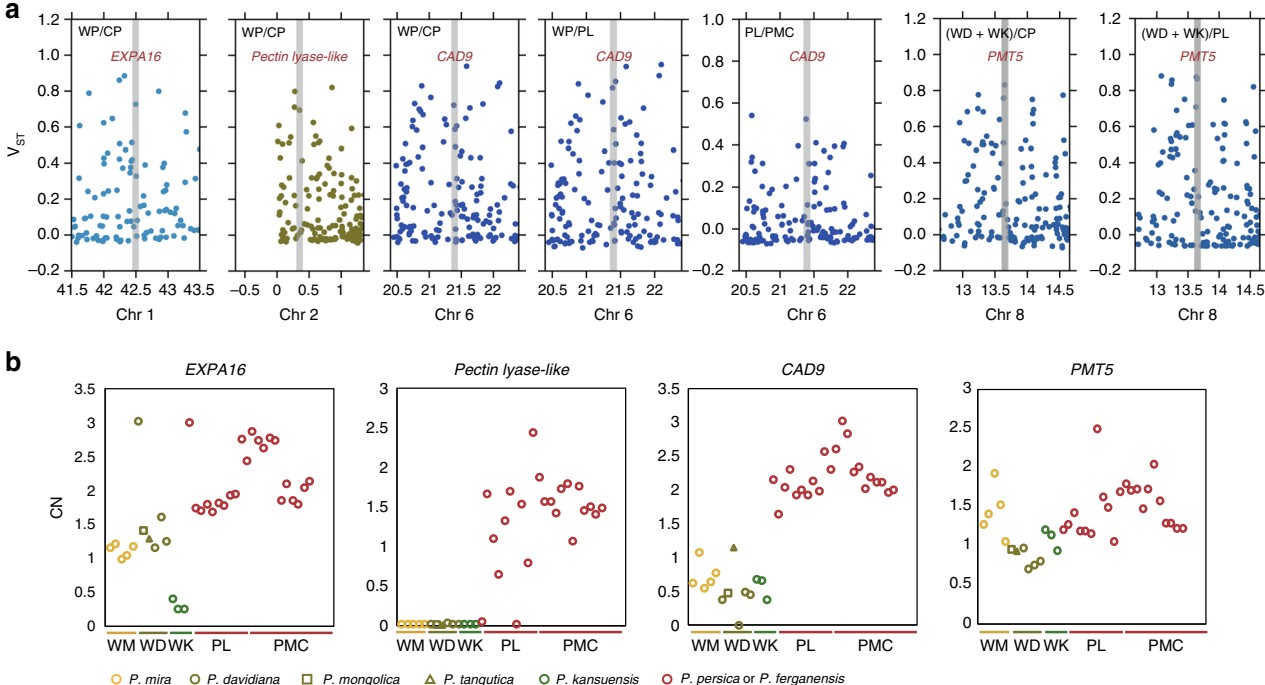

**Fig. 5** CNVs involved in fruit texture and taste during peach domestication and improvement. **a** Manhattan plots of $V_{ST}$ values. Four fruit texture or taste associated candidate genes (*EXPA16*, *Pectin lyase-like superfamily protein*, *CAD9*, and *PMT5*) (within regions exhibiting high $V_{ST}$ values (top 5%)) are visualized with shadows. **b** Copy number profiles of the four candidate genes with high $V_{ST}$ values. The selected genes and estimated copy numbers for individuals from distinct groupings are provided in each plot

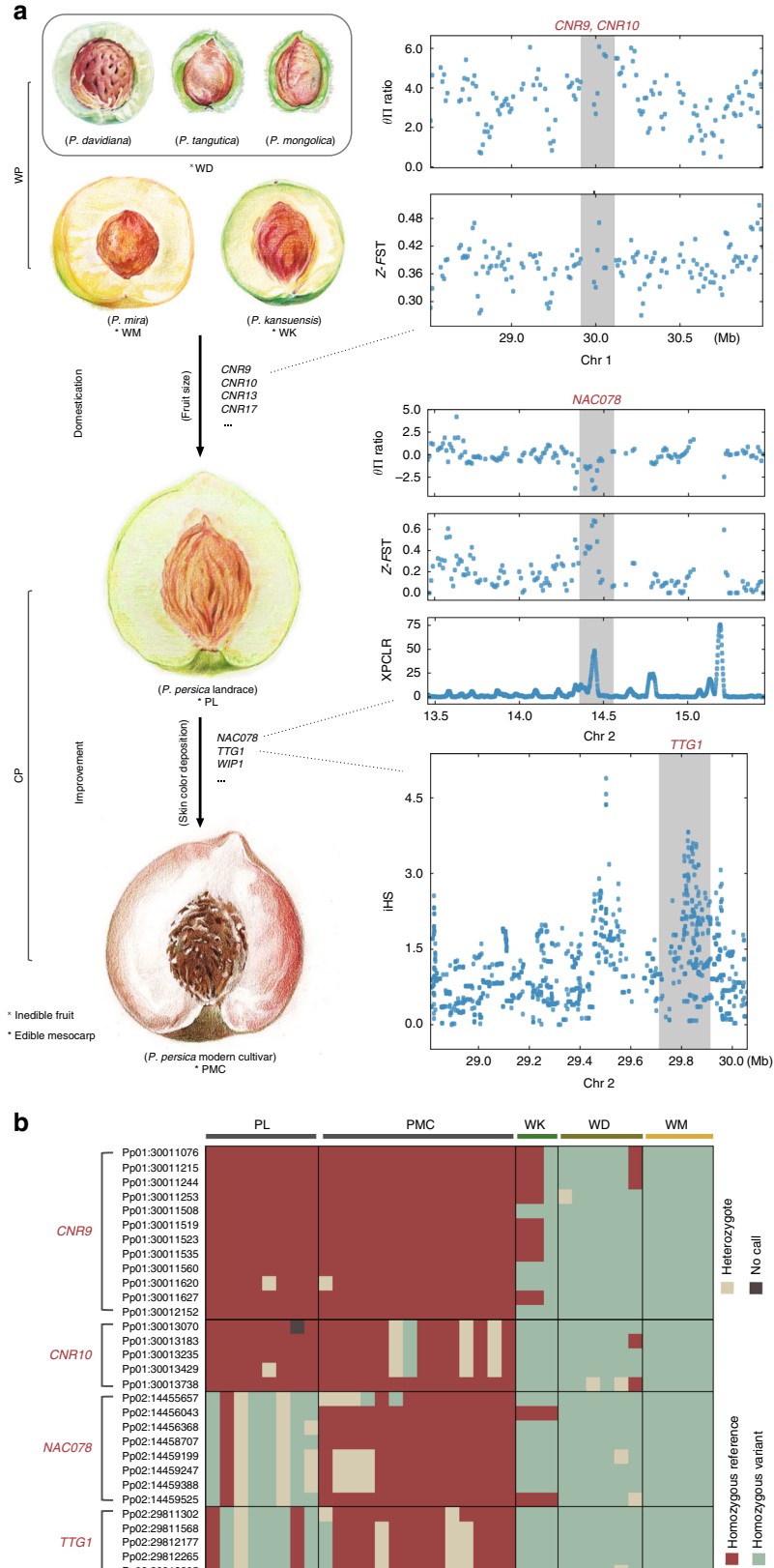

**Fig. 6** Stage-wise selection for fruit size and skin color during peach domestication and improvement. **a** Schematics of fruit evolution during peach domestication. Several candidate genes underlying peach fruit size and skin color were shown, and of these genes, *CNR9*, *CNR10*, *NAC078*, and *TTG1* within regions having positive selection signatures were visualized with gray shadows in Manhattan plots. Fruit images were created by Yang Yu and Shan Gao. **b** Haplotype differentiation patterns of *CNR9*, *CNR10*, *NAC078*, and *TTG1*. Across the PL, PMC, WK, WD, and WM accessions, significant haplotype differentiation patterns were observed for several SNPs within these several candidate genes related to fruit size (*CNR9* and *CNR10*) and skin color (*NAC078* and *TTG1*). The haplotype differentiation patterns were further confirmed using the peach genome data from Cao et al.[8] (Supplementary Fig. 16)

Fig. 11 and Supplementary Data 11 and 17) in the CP group, of which *PpCNR13* overlaps with a reported fruit weight QTL in the peach genome[45]. Moreover, three *CNR* genes (*PpCNR9*, *PpCNR10*, and *PpCNR17*) (Supplementary Fig. 10 and Fig. 11 and Supplementary Data 11 and 17) were also detected in the PL subgroup vs. the WP group and vs. a group containing both the WD and WK accessions (Fig. 6a). Haplotype differentiation patterns of several *PpCNR9* and *PpCNR10* loci clearly separated the WP accessions from the CP accessions (Fig. 6b), suggesting their potential contributions to increased fruit size.

We also identified genes that are known to influence the color of various fruits[46–48] (orthologs of *NAC078*, *WIP1*, and *TTG1*) in a comparison between the PL subgroup (less skin color depositing) and the PMC subgroup (more skin color depositing) (Supplementary Fig. 15 and Supplementary Data 11 and 18). The haplotype differentiation patterns of several *NAC078* and *TTG1* loci separated the PMC accessions from both the PL accessions and the WP accessions (Fig. 6b), suggesting that peach fruit skin color has been selected recently in the course of peach improvement. We also found similar haplotype differentiation patterns for *CNR* and skin color related genes when using additional genome data from Cao et al.[8] (Supplementary Fig. 16), further supporting the evolutionary history in peach wherein selection for fruit size occurred during peach domestication prior to a subsequent selection for fruit skin color.

## Discussion

This study examined the evolutionary history of fruit edibility in peach by combining genomic analyses with paleo-geographic data and archeological evidence. Our results suggest that topographical changes caused by the uplifts of Tibetan plateau had driven the diversification of several peach species during the Pliocene. These uplifts made southwest China a favorable region for the emergence of fruit edibility in peach species and the origin of *P. persica*. It has been suggested that humid environments offer desirable conditions for the emergence of fleshy mesocarps[7]. We here present evidence supporting that humid southwest China provided not only the glacial refugia for peach species, but also fostered enhanced frugivore-plant interactions for the emergence of fleshy fruit. Our study also provides molecular evidence to support the previously proposed idea that frugivore-mediated selection can drive the evolution of fruit traits[30].

As the ancestor of *Amygdalus* and *Persica* sections' species likely had dry, splitting inedible mesocarps, it has been suggested that the transition to non-splitting, fleshy mesocarp that occurred in *Persica* section species likely result from selection and domestication[6,7]. Specifically, a phylogenetic analysis given by Delplancke et al.[18] showed that *Persica* section species diverge into two subclades, including one subclade covering inedible peach species (*P. davidiana*, *P. mongolica*, and *P. tangutica*) and one subclade covering fleshy peach species (*P. mira*, *P. kansuensis*, and *P. persica*)[18] (Supplementary Note 4). Viewed alongside a peach genome study from Cao et al.[8], our results add more resolution to clarify that *P. mira* was the earliest diverged wild relative species of *Persica* section, followed by the inedible subclade (*P. davidiana*, *P. mongolica*, and *P. tangutica*), *P. kansuensis*, and *P. persica*. The emergence of fruit edibility likely occurred in the common ancestor of *P. kansuensis* and *P. persica* circa 3.14–2.6 Mya. While fruit edibility in *P. mira* was probably acquired later from an ancient introgression from the common ancestor of *P. kansuensis* and *P. persica*. These results in peach expand our general understanding of introgression that can cause the delivery of domestication syndrome between wild species and domesticates.

The ancestor of *Amygdalus* subg. species was likely self-incompatible[6], and we proposed that the increased inbreeding of the three edible peach species possibly resulted from selection. We suspect that the maintenance of edible fruit traits was aided along by their high inbreeding levels (Supplementary Note 5), and this idea is reinforced by the fact that the self-compatibility has been shown to cause the stabilization of peach varieties. Assuming selection is occurring, the requirement of maintaining fruit traits (those desired by frugivores) could have increased the inbreeding level in the populations of the edible peach species (Supplementary Note 2), reflecting an evolutionary relation between these two important domestication syndromes. Therefore, the increased extent of ROHs, and *F* values for the genomes of the edible species could also be seen to reflect selection footprints of the edible peach species (Fig. 1c). Finally, we raised that *P. persica* has experienced profound domestication and improvement processes, while *P. mira* and *P. kansuenisis* are staying at pre-domestication stage (Supplementary Note 6), a transition phase from wild to domesticates.

Focusing on *P. persica*, the completely domesticated species in *Persica* section, we scanned the genome for selection signatures underlying fruit texture, taste, size, and skin color, and identified a large set of SNPs and candidate genes associated with these fruit traits. Our results also highlight the contribution of CNVs that have impacted quantitative traits for fruit texture and taste during peach domestication and improvement. Similar to the scenarios that occurred during the domestication of tomato[44] and apple[42], *FW2.2*/*CNR* family genes likely contributed to the increase of fruit size during peach domestication. In contrast, genes related to fruit skin color were selected recently in the course of improvement that has led to peach modern cultivars. Our study dramatically increases the amount of genomic data available for peach, provides valuable information for facilitating marker-assisted selection, and clarifies the evolutionary history of specific fruit traits in peach, offering a new evolutionary model that help us to understand the evolution of perennial fruit tree crops.

## Methods

**Sample sequencing**. All the 44 accessions newly sequenced in this study are listed in (Supplementary Table 2). Genomic DNA was extracted from leaf samples using Plant DNA Mini Kits (Aidlab Biotech, China). 1.5 μg of high quality DNA per sample was used to prepare the libraries. Sequencing libraries were generated using a Truseq Nano DNA HT Sample Preparation Kit (Illumina, USA) following the manufacturer's recommendations; index codes were added to each sample. The insert size of each library was ~350 bp. The quality and quantity of libraries were analyzed using an Agilent 2100 Bioanalyzer instrument and qPCR. Whole genome paired-end reads were generated using Illumina platforms (HiSeq2000/HiSeq2500/HiSeq X Ten).

**Read mapping and SNP calling**. The high quality paired-end reads were mapped to the *P. persica* Genome v2.1.a1 (http://www.rosaceae.org/species/prunus_persica/genome_v2.0.a1) reference genome using BWA[49] (v0.7.12) with the parameter: 'mem -t 4 -k 32 –M'. PCR or optical duplicates were removed using SAMtools[50]. We performed SNP calling using a HaplotypeCaller approach as implemented in the package GATK[51] (Genome Analysis Toolkit, v3.3-0-g37228af). To remove the potential false positive SNPs, SNPs with QD<2.0 or FS>60.0 or MQ<40.0 or MQRankSum<−12.5 or ReadPosRankSum<−8.0 were filtered. The accuracy of called SNPs was assessed using the customized Axiom® Peach 170 K SNP array with SNPs from 96 cultivated peach accessions (including the cultivated peach accessions in this study and from two previous studies;[8,10] Supplementary Data 1). As the SNP array was developed based on the *P. persica* Whole Genome v1.0 reference genome, a liftover program (http://genomewiki.ucsc.edu/index.php/Same_species_lift_over_construction) was applied to transit the genotyped SNPs from the *P. persica* Whole Genome v1.0 reference genome to the *P. persica* Whole Genome v2.1.a1 for SNP validation.

**Annotation of genetic variants**. Gene-based SNP annotation was performed according to the annotation v2.1.a1 of the *P. persica* genome using the package ANNOVAR[52] (v2013-06-21). Based on the genome annotation, SNPs were categorized as occurring in exonic regions (overlapping with a coding exon), intronic regions (overlapping with an intron), splicing sites (within 2 bp of a splicing

junction), upstream and downstream regions (within a 1 kb region upstream or downstream from the transcription start site), or intergenic regions. SNPs in coding exons were further grouped as either synonymous SNPs or nonsynonymous SNPs. Additionally, mutations causing gain of stop codon and loss of stop codon mutations were also classified as nonsynonymous SNPs.

**Population structure**. To analyze the population structure we screened a subset of bi-allelic and high quality SNPs with a call rate ≥90% and a minor allele frequency ≥5%. A neighbor-joining tree was constructed using the program TreeBeST (v1.92) (see URLs) with 100 bootstrap replicates. The tree was displayed using MEGA7[53] and FigTree (v1.4.2) (see URLs). To infer the population structure, we used ADMIXTURE[12] (v1.23), which implements a block-relaxation algorithm. To make consideration for HWE violations, we filtered SNPs by testing HWE violations ($P > 10^{-4}$) and reconstructed the model-based clustering analysis. To identify the best genetic clusters K, cross-validation error was tested for each K value from 2 to 15. The termination criterion was $10^{-6}$ (stopping when the log-likelihood increased by less than $\varepsilon = 10^{-6}$ between iterations). We also performed principal component analysis (PCA) using the program flashpca[54] (v1.2.6).

**Median joining network**. A long haplotype (98.32 kb) was used to build Median Joining Networks with the software PopART[55], with default parameters. Strongly linked blocks were detected using PLINK, with default parameters. SNPs in these blocks were phased by BEAGLE (r1399)[56].

**Identity score (IS) and identity-by-state (IBS)**. To assess the similarity between two accessions, pairwise IS and IBS similarity values were calculated. The IS at each SNP was calculated using the formula $IS=1-(Ra-Rb)$, where Ra and Rb are the ratio of reads supporting reference in, respectively, sample a and b. Genome-wide average pairwise IBS similarity were calculated using PLINK (v1.07)[57].

**Genetic diversity**. Genetic nucleotide diversity ($\theta\pi$, the average number of nucleotide differences per site between two randomly chosen DNA sequences from the population), Tajima's D, and population-differentiation statistics (Fixation index, $F_{ST}$) were calculated using VCFtools[58] (v0.1.14) with a 20 kb sliding window with a step size of 10 kb across the peach genome.

**Regions of homozygosity**. To compute runs of regions of homozygosity (ROH), we first pruned the SNPs by LD and removed low frequency markers using PLINK[57] with the following parameters: '–indep 50 5 2 –maf 0.05'. We computed ROHs with a sliding window of 50 SNPs (–homozyg-window-snp 50), allowed 2 missing calls (–homozyg-window-missing 2), and did not allow heterozygotes (–homozyg-window-het 0) in the window. The final ROHs (i) contained at least 50 SNPs (–homozyg-snp 50), (ii) were longer than 50 kb (–homozyg-kb 50), and (iii) had a minimum "SNP density" of 50 (–homozyg-density 50).

**Inbreeding coefficient**. The coefficient of inbreeding (F value) in each population was calculated using the formula $F=1-Ho/He$, where Ho is the observed heterozygosity and He is the expected heterozygosity.

**Linkage disequilibrium**. To estimate and compare the pattern of linkage disequilibrium (LD) of each population, the squared correlation coefficient ($r^2$) values between any two SNPs within 500 kb were computed by using the software Haploview[59] (v4.269). To get reliable results, all of the accessions (Supplementary Data 2) sequenced in this study and data for accessions downloaded from two previous studies[8,10] were included in this analysis. We randomly sampled 5 accessions in each sample for 100 times to eliminate the bias of sample size. We produced an LD decay plot that shows the average $r^2$ values in a bin of 100 bp against the physical distance of pairwise bins. The standard deviations were derived from the 100× resampling.

**Divergence time**. The divergence times of P. persica, P. kansuensis, P. davidiana, P. mira, and P. dulcis were estimated using BEAST2[60], with a substitution rate of $7.7 \times 10^{-9}$ per site per generation and a generation time of 7 years. The substitution model of HKY was selected, and the mutation frequencies were estimated internally. All 4-fold degenerate sites were used as a molecular clock, because these sites were considered as neutral evolution. To avoid ascertainment bias from the use of P. persica as a reference, all reads with mapping quality <20 were removed, and all sites having a coverage depth below 15× were masked. We selected one accession as representative from each species. At heterozygous sites, one allele was randomly selected. In total, 761 complete genotyped genes were selected and a total of 140,557 degenerate sites were used to estimate divergence time. The same procedure was also used to estimate the time of the most recent common ancestor of cultivated P. persica.

**Demographic history**. MSMC2[61] was used to infer demographic history. To improve reliability, genome regions were masked when the coverage depth was below 15× after removing reads with mapping quality <20 and were masked using

Heng Li's SNPable tool (see URLs). First, we split the reference genome into overlapping 35-mers and then mapped these back to the reference genome using BWA (bwa aln -R 1000000 -O 3 -E 3). Only regions of the majority of overlapping 35-mers were mapped back uniquely and without 1-mismatch were kept. The maximum 8 haplotypes, except P. kansuensis which was used all 6 haplotypes were used in each population. Scaled times were converted to generation, assuming a generation time of 7 years (full reproductive age) and a mutation rate of $7.7 \times 10^{-9}$ per site per generation[17]. The atmospheric surface air temperature relative to the present (°C) and the ice volume contribution to the marine isotope signal (relative to the present) were downloaded from the NCDC database (see URLs).

**Introgression**. TreeMix[22] (v1.12) was used to model gene flow in peaches. This method first infers a maximum likelihood tree based on genome-wide allele frequency data and then identifies potential gene flow from a residual covariance matrix. P. dulcis was used as the outgroup. To account for linkage disequilibrium between adjacent SNPs, we grouped them together in windows of 100 SNPs ('-k 100'). The number of migration events ('-m') was modeled from 0 to 4. We also did a round of global rearrangements after adding all populations ('-global'). We used Patterson's D-statistic (ABBA-BABA test), which was implemented in the software package ADMIXTOOLS (v1.0.1)[62] to test introgressions between P. kansuensis and P. mira, and between P. persica and P. mira, with P. dulcis as the outgroup. The significance was assessed by a block jackknife procedure and a block size of 5 Mb was used. The D-statistic and a modified f-statistic ($f_d$)[63] were integrated to identify the introgression region with a sliding window of 20 kb and a step size of 10 kb across the genome. Windows with SNPs <30 or negative D-statistic values were excluded[63]. The windows with the top 5% of $f_d$ values were considered as candidate introgression regions. Mean pairwise sequence divergence ($d_{XY}$) was used to distinguish introgression and ancestral structure[63]. The introgression times were estimated using BEAST2 as described above with the 4-fold degenerate sites of the genes in three longest (24.5/24.2/23.7 Kb) strongly linked blocks in the introgressed segments.

**Selective sweep**. We used multiple methods to detect regions and genes under selection. SNPs with minor allele frequency below 5% were removed from this analysis. To identify potential selective sweeps between population A and population B, $\log_2(\pi B/\pi A)$ and $F_{ST}$ were calculated together using VCFtools with a 20 kb sliding window and a step size of 10 kb. Windows that contained less than 10 SNPs were excluded from further analysis. The windows that were simultaneously (1) in the top 5% of Z-transformed $F_{ST}$ values and (2) in the bottom 5% $\log_2(\pi B/\pi A)$ were considered to be candidate selective regions in population A. XP-CLR is a method that uses allele frequency differentiation at linked loci between two populations to detect selective sweeps. Each chromosome was analyzed using the program XP-CLR[64] (v1.0) with parameters '-w1 0.0005 200 200 1 -p1 0.9'. The average XP-CLR scores were calculated for each 20 kb sliding window with a step size of 2 kb. The windows with the highest 1% of XP-CLR scores were considered as candidate selective regions. XP-EHH[23] and iHS[65] were implemented using the program Selscan (v1.1.0)[66]. Results were normalized with a 20 kb window. The ratio of extreme scores (|score| > = 2) in each window were calculated. The top 1% of windows (with the highest ratio of extreme scores) were considered to be candidate selective regions. The results each of the above methods were combined. The genes in the merged candidate selective regions along the peach genome were considered as candidate selective genes. Gene Ontology (GO) enrichment analysis of selective genes was implemented with the GOseq[67] R package. GO terms with corrected P-values <0.05 were considered significantly enriched.

**CNVs calling and evaluation of potential selection of CNVs**. Copy Number Variations (CNVs) were identified by CNVnator (v0.2.7)[68]. This method is based on read depth and using refining based on GC correction. Samples were called with a bin size of 1 kb; the other parameters were the defaults. Fragments that passed the criteria (mean depth <0.5 or >1.5) were kept as CNVs. Candidate CNVs were further filtered for N regions and repetitive regions of the reference genome using BEDtools. For CNV select using CNVs data, we first used a previously reported method to convert the CNVs values into genotypes for individual CNV fragments (CNVRs);[69] then, the genotypes for individual CNVRs were used for CNV identification. To explore population differentiation using all CNVs, we devised a statistic, $V_{ST}$, that estimates population differentiation based on the quantitative intensity data and varies from 0 to 1; similar to $F_{ST}$, $V_{ST}$ is calculated by considering $(V_T-V_S)/V_T$ where $V_T$ is the variance in log2 ratios apparent among all unrelated individuals and $V_S$ is the average variance within each population, weighted for population size. Estimated CN to a given genomic region for each individual was calculated using the CNV genotyping procedure in CNVnator (v0.2.7).

**URLs**. Global climate: https://www.ncdc.noaa.gov
TreeBeST: http://treesoft.sourceforge.net/treebest.shtml
FigTree: http://tree.bio.ed.ac.uk/software/figtree/
SNPable tool: http://lh3lh3.users.sourceforge.net/snpable.shtml

## Data availability

All relevant data are available from the authors. The raw sequence reads have been deposited into the NCBI Sequence Read Archive under project PRJNA310042. A reporting summary for this Article is available as a Supplementary Information file.

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

## Acknowledgements

We acknowledge the help of Beijing Agro-biotechnology Research Center and Beijing Academy of Forestry and Pomology Sciences, Beijing Academy of Agricultural and Forestry Sciences, and Zhengzhou Fruit Research Institute, Chinese Academy of Agricultural Sciences for the assistance in sample collection. We would like to thank Prof. Scott Jackson (University of Georgia), Prof. Yuehua Cui (Michigan State University), and Prof. Jun Yu (Beijing Institute of genome research, Chinese Academy of Sciences) for manuscript revision and kind advice on this manuscript. We would also like to thank Beijing Municipal Science & Technology Commission for supporting this research through Beijing Municipal Science & Technology Project (Z151100001015005) to Hua Xie, and Beijing Academy of Agricultural and Forestry Sciences for supporting this research through Science & Technology Innovation Project (KJCX20170203) to Jianhua Wei.

## Author contributions

Hua Xie, Jianhua Wei, Quan Jiang, Rongcai Ma, Hongwei Zhao, and Shilin Tian designed and managed the project. Hua Xie, Yang Yu, Jun Fu, Hongwei Zhao, and Shilin Tian drafted and revised the manuscript. Jun Fu, Yang Yu, Yaoguang Xu, Jiewei Zhang, Fei Ren, Wei Guo, Xiaolong Tu, Jing Zhao, Dawei Jiang, Weiying Wu, and Gaochao Wang performed the analyses. Hua Xie, Jiewei Zhang, Fei Ren, and Jianbo Zhao collected all the samples.

## Additional information

**Competing interests:** The authors declare no competing interests.

