## [Peer Review File · Nature Communications]

Reviewer #1 (Remarks to the Author):

This manuscript describes the history, evolution and domestication of peach through analyzing the deep genome resequencing data of a wide range of cultivated and wild peach species as well as other related species such as cultivated almond. The authors provided genetic and molecular evidence to support that fruit edibility of peach had emerged long before the artificial selection process, mainly contributed by frugivore-mediated selection, and then further improved during the domestication process. The manuscript provided some novel findings regarding peach evolution as well as useful genomic resources for peach breeding and research.

Major concerns

1. I found the first several pages of the manuscript is very hard to understand due to the lack of sufficient background information. Examples are given below:

a) Lines 26 and 28: mentioning *P. kansuensis* and *P. mira* without providing any background information could cause difficulty in understanding for most of the readers who don't have much peach taxonomy knowledge.

b) Line 58: it's very hard to follow "Amygdalus species" here without any background information provided.

c) Line 76-78: Need to provide background information on the selected species and why they were selected, as well as a summary on the number of accessions in each species and which species have edible fruits. I am curious why the author included cultivated almond, *P. pedunculata* and *P. triloba* in the analysis.

2. One major conclusion, as stated in the abstract, is "fruit edibility was acquired by *P. mira* from an ancient introgression from this common ancestor". Beside the Treemix analysis, I would suggest addition analysis to obtain more evidence to support the direction of the introgression. The D-statistic the authors used can only detect introgressions, not the direction. The authors should look at partitioned D-statistic described in Eaton and Ree (2013. Inferring phylogeny and introgression using genomic RADseq data: An example from flowering plants (Pedicularis: Orobanchaceae). Systematic Biology 62: 689), which does allow for the inference of direction of introgression.

3. I am concerned about the choose of wild species used in the selective sweep analysis. The authors used all wild peaches (WP, line 252) including *P. mira*, *P. kansuensis*, *P. mongolica*, *P. tangutica*, and *P. davidiana* to identify selective signatures in the peach genome. This would complicate the identified signatures as some of them are not related to domestication but rather population differentiation, since some of the wild species are not involved in the domestication process at all. I understand that the direct wild progenitor of cultivated peaches remains unclear, but some species which are believed not to be the progenitor (e.g., *P. mira* based on this study?) should be excluded in the analysis to reduce the noises in the results.

Minor concerns:

4. "peach (Rosaceae)" in the title is misleading. Rosaceae is a family that peach belongs to.
5. Line 21: change "Peach is" to "Peach (*Prunus persica* L. Batsch) is".
6. Line 42, "estimated to 265 Mb": should be "estimated haploid genome size of 265 Mb".
7. Line 52-57: I couldn't follow the logic here. How "indistinguishable endocarp" can suggest "edible fruits" (mesocarp)?
8. Line 60-61: I don't think fruit size and skin color are fruit edibility traits.
9. Line 78-84: Supplementary Figure 1 is not needed. It's largely duplicated with Supplementary Table 2 and 4. Supplementary Table 3 should be removed. It's not necessary. Just need to cite the genome paper or the link to the genome sequence. It's very strange that in Supplementary Table 5 the numbers in the "Synonymous" column are almost all 15,138.
10. Line 100-103: need to cite the corresponding figure for this result.
11. Line 130, "among these five peach species": Is almond (*P. dulcis*) a peach species?
12. Line 158: change to "which is comparable to the size of modern peaches while remarkably larger than the fruit of extant non-edible peach species."
13. Line 160-162: How the "0.60 Mya" was derived?
14. Line 168 and 175: I suggest to mark the two periods (115-130 Kya and 19.0-26.5 Kya) on Figure 2b.
15. Line 193: change "The early" to "Early"
16. Line 194: change "this that" to "this".
17. Line 348: change "Reads mapping" to "Read mapping"
18. Line 351: Provide a reference for SAMtools
19. Line 355: Change "SNPs annotation" to "SNP annotation"
20. Line 376: Change "reads ratio" to "ratio of reads"
21. Line 402-403: How the "substitution rate of 7.7×10^{-9} per site per generation and a generation time of 7 years" was derived?
22. Line 428-429, "we assumed 1 cM was equal to 1 M bp": this seems to be a wrong assumption. Can the authors provide data or references to support this assumption?

Reviewer #2 (Remarks to the Author):

The manuscript of Yu et al. (NCOMMS-18-05001) proposes a model for evolution of cultivated peach and related species from its origin in southwest China. The research combines genetic analysis of resequencing data from an ample collection of accessions including mume, almond, various peach wild relatives and cultivated varieties, with archaeological information, providing evidence for several highly relevant aspects of peach history and evolution, some of the most important being : 1) The establishment of the relationships between the group of *Prunus* species analyzed and the identification of introgression patterns between some of them; 2) the sequence of emergence of the different species, and their demographic story in relationship with the geological and climatic events of the different steps of evolution; 3) the evidence that fruit edibility was selected prior to domestication, possibly involving frugivores; 4) the indication that fruit size was selected earlier after domestication and selection for fruit color is a much more recent event. Overall the paper provides a well-supported and singular evolutionary process for peach, different from that of other well studied perennial species such as apple and grape, and that may represent by itself a model for other tree or herbaceous crops. In addition, the paper provides useful information on candidate genes involved in the inheritance of key economical characters of interest for geneticists and breeders.

My view is that the manuscript describes excellent original research and is acceptable for publication in Nature Communications. I have only two comments and a few minor suggestions for improvement that are described below.

An interesting suggestion of the manuscript is that monkeys may have played a role in the selection for fruit edibility in peach and *Prunus mira* before domestication. As evidence for this, the N_e trajectories of three species during the Pleistocene are provided, two of which possibly sharing the same habitat with peach and another species that did not. The authors consider that the two former species shared similar trajectories but not the third, which is something that was not evident to me just by looking at Figure 2. A more detailed explanation or a statistical test supporting this trajectory similarity would help to make this stronger evidence.

The authors identify a number of putative selection sweeps comparing the wild, landrace and modern cultivar compartments, and then find in these regions genes involved in fruit characteristics of agricultural interest. Some relevant examples are provided suggesting selection for genes known to affect these characters in other species that are located in the selection sweeps. This is a reasonable approach, but one could argue that provided a sufficiently large number of candidate genes, the probability of finding one or several of these genes by chance in the genome fragments identified is high. Would it be possible for one or more of these characters to estimate the total number of genes possibly involved and to test whether the numbers of genes identified are significantly higher than those expected by chance?

Minor changes proposed:

Abstract

Line 23: change "... species are closely..." to species "... species is closely..."

Line 31: change "...during peach..." to "...during modern peach..."

Introduction

Line 72. "...our novel hypothesis...". This hypothesis was already proposed in the paper of Su et al. (2015), cited in the manuscript.

Results

Line 97. Change "...maturity and morphologically..." to "...maturity and are morphologically..."

Line 194. Change "...and this that..." to "...and this..."

Line 233. Change "...the basis characteristics..." to "...the basic characteristics ..."

Lines 283-284. A reference could be provided to support the statement that stone and fruit size are strongly correlated.

Methods

Line 421. Change "...based genome-wide..." to "...based on genome-wide..."

Figure Legends

Line 676. Change "...Linage..." to "...Linkage..."

Supplementary Figure 6. The *P. ferganensis* group appears to be the one marked with dark green color, but it is not sufficiently clear. It could be stated on the legend or indicated on the diagram.

Reviewer #3 (Remarks to the Author):

The authors used 58 high-coverage genomes of cultivated peach and closely related relatives to explore the evolutionary history, including the genetic divergence of cultivated peaches and its closely related relatives, demographic trajectories, domestication features and their evolved time. The topics addressed are interesting, but there are several problems with the manuscript as presented.

The exploration of data analysis with appropriate tools is quite weak and some of the conclusions are lack of evidences to support. There are several issues with the manuscript that need to be addressed before the manuscript is published.

Major issues:

1. SNP calling and filtering is not clear enough. The author reported that the total SNPs called are 24,280,369, but it is unclear why only use 2,559,589 SNPs to do PCA and 3,909,617 to build NJ tree and how they did filtering. Please also clarify how the author did SNP validation with SNP array data and how many of those SNPs can be found in the resequencing data. In order to see if the SNPs called from whole genome resequencing data are reasonable, please show the site frequency spectrum (SFS) and the ratio of transition and transversion (Ts:Tv). Since the author did annotations for all of the SNPs with Annovar, but it is unclear how many of those SNPs are noncoding, coding, synonymous or nonsynonymous SNPs.
2. Some of the analyses applied are either potentially inappropriate or over interpreted. For example, the authors make use of ADMIXTURE for population structure. However, ADMIXTURE likely makes a Hardy Weinberg assumption regarding expected genotypes. Violations of this assumption could lead to over clustering of samples owing to inbreeding. Did the authors have a strategy for addressing this in samples that are highly inbred?
3. The authors used VCFtools to calculate genetic nucleotide diversity, Tajima's D and Fst. VCFtools were designed to calculate the population genetic statistics for diploid samples, but the majority of the samples in this study are highly selfing (which will be treated as haploid), and can not VCFtools to calculate genetic nucleotide diversity, Tajima's D and Fst.
4. All of the scripts and analytical tools used for a largely computational project like the work reported here need to be made available through Github, Bitbucket, or a similar version control repository. Because the scripts are essential to reproducing this work, they should be available to reviewers and/or readers. These resources are important in part because samtools, GATK, and similar utilities include a large number of options that alter the resulting output.
5. Some of the arguments or conclusions seem to be lack of the evidences to support. For example, in Line 129, the authors argued that "it appears that inbreeding level has likely played a major role in determining the level of LD among these five peach species", but apparently the authors did not examine the influences of other factors except inbreeding. Another example is in line 237 to line 245, the authors proposed that " the ancient selection process likely involved frugivorous, contributing to the emergence of fruit edibility, prior to recent human-mediated domestication" only based on that the Ne trajectories of three monkey species has the similar pattern with the peaches in this study.
6. The author argued that " it is reasonable to infer that common ancestor of *P.persica* and *P.kansuensis* was self-incompatible. We therefore propose that the combination-incompatibility and overlapping habits...approximately 3 million years ago". Is *P.kansuensis* self-incompatible? Why is it reasonable to infer that common ancestor of *P.persica* and *P.kansuensis* was self-incompatible? Again it is hard to follow the statement that "We therefore propose that the combination-incompatibility and overlapping habits...approximately 3 million years ago". How did the author get this conclusion?

7. Line 221-223, how did the author know stronger signatures in introgressed segments compared with genome levels of *P. mira*? Based on which comparison?

Minor issues:

1. Line 47, please fix the citation format.

2. Line 114, "An increase in the ratio of nonsynonymous to synonymous SNPs.." in line 114, did the author use the private SNPs to do calculation?

3. Which tools did the author to calculate the ROH? Please clarify in the Method.

4. Which tool did the authors calculate the inbreeding coefficient? Please clarify in the Method.

5. Please cite the paper for the substitution rate of 7.7×10^{-9} in the main text.

6. Please label the samples consistently. For example, use those labels CA, WM, WD, WK, CP in all of the figures, tables, and supplemental tables, otherwise it is hard to follow.

Response to Reviewers' comments

Reviewer #1 (Remarks to the Author):

This manuscript describes the history, evolution and domestication of peach through analyzing the deep genome resequencing data of a wide range of cultivated and wild peach species as well as other related species such as cultivated almond. The authors provided genetic and molecular evidence to support that fruit edibility of peach had emerged long before the artificial selection process, mainly contributed by frugivore-mediated selection, and then further improved during the domestication process. The manuscript provided some novel findings regarding peach evolution as well as useful genomic resources for peach breeding and research.

Major concerns

1. I found the first several pages of the manuscript is very hard to understand due to the lack of sufficient background information. Examples are given below:

a) Lines 26 and 28: mentioning *P. kansuensis* and *P. mira* without providing any background information could cause difficulty in understanding for most of the readers who don't have much peach taxonomy knowledge.

b) Line 58: it's very hard to follow "Amygdalus species" here without any background information provided.

c) Line 76-78: Need to provide background information on the selected species and why they were selected, as well as a summary on the number of accessions in each species and which species have edible fruits. I am curious why the author included cultivated almond, *P. pedunculata* and *P. triloba* in the analysis.

Response:

Thanks for the comments and helpful suggestions. To address this issue, we have added some background information about peach taxonomy to help set the stage for the results and ideas described in the manuscript.

For answering questions **a)** and **b)**, we put the description in Line 53–57:

"*Prunus* subgenus *Amygdalus* comprises two sections, *Amygdalus* and *Persica*, in which domesticated almond and peach are classified, respectively. The common ancestor of this subgenus likely bears a dry, splitting mesocarp, while the transition to fleshy and palatable mesocarp in some *Persica* section species (wild *P. mira* and *P. kansuensis*, and *P. persica*) (Supplementary Table 1) was presumably derived from selection and domestication in China^{8,9},"

For answering the question **c)**, we put the description in Line 78–75:

"Previous studies support that peach was domesticated in China⁴, while the emerged fleshy mesocarp phenotypes are found not only in *P. persica* but also in *P. mira* and *P. kansuensis*."

These facts suggested a way for us to explore the selection and/or domestication history that has driven the transition to fleshy edible fruit in some *Persica* section species. In order to clarify the taxonomy of the subg. *Amygdalus* species endemic to China and to study the evolved fruit edibility in peach species, we generated a total of 700 Gb of raw sequence data for 44 high-coverage genomes of cultivated peach and all of the known wild relative species (belonging to *Persica* section) endemic to China, and its closely related species (Supplementary Table 2).”

A summary of the number of accessions in each species and their morphological characters for each species (*e.g.*, edible or inedible fruits, skin color, *etc.*) is provided in Supplementary Tables 1 and Table 4. As noted in our revised text quoted above, our panel included cultivated peach and all of the wild relative species (belong to *Persica* section) endemic to China, as well as cultivated almond (belongs to *Amygdalus* section) and three other closely related relatives (Supplementary Table 2). All of these closely related relative species may help us to understand the evolution of peach. More specifically, *P. mongolica* and *P. tangutica* have been recognized as *Amygdalus* section species; they are closely related to almond (*P. dulcis*). As you may be aware, Charles Darwin actually proposed that peaches are almonds that have been modified in a marvelous manner¹. Our results provide strong molecular evidence against that theory and clarify several other important issues about the evolutionary process of peach species. In addition to the fact that they are prevalent in China, *P. pedunculata* and *P. trilob* have been classified into to a sister clade of the peach clade in a phylogenetic analysis based on morphological data², so we also included these two species in current study and mentioned these issues in the revised manuscript.

References

- Darwin, C.R. The variation of animals and plants under domestication, 2nd edn. John Murray, New York, pp 357–441 (1868).
- Yazbek, M. & Oh, S.H. Peaches and almonds: phylogeny of *Prunus* subg. *Amygdalus* (Rosaceae) based on DNA sequences and morphology. *Plant Syst. Evol.* **299**, 1403–1418 (2013).

2. One major conclusion, as stated in the abstract, is “fruit edibility was acquired by *P. mira* from an ancient introgression from this common ancestor”. Beside the Treemix analysis, I would suggest addition analysis to obtain more evidence to support the direction of the introgression. The D-statistic the authors used can only detect introgressions, not the direction. The authors should look at partitioned D-statistic described in Eaton and Ree (2013. Inferring phylogeny and introgression using genomic RADseq data: An example from flowering plants (Pedicularis: Orobanchaceae). *Systematic Biology* 62: 689), which does allow for the inference of direction of introgression.

Response:

Thanks for this suggestion about the introgression analysis in our manuscript. Our study of the introgression among peach species was conducted by using both the *D*-statistic and TreeMix model in our manuscript.

Following the suggestion, we have tried to use partitioned *D*-statistic test described in Eaton and Ree (2013. Inferring phylogeny and introgression using genomic RADseq data: An example from flowering plants (*Pedicularis*: Orobanchaceae). Systematic Biology 62: 689) to infer the gene flow. However, we found that although it is a good way to infer gene flow direction, the proposed approach does not work technically when applied to our current study due to the unfit phylogenetic relationship. This analysis need a five-group phylogenetic relationship (showed as below): O is the out group, P1 and P2, and P3 and P4, are divergent from their common ancestor P12 and P34 respectively. Like the event showed in this figure, a gene flow from P3 to P2 occurred. Thus we can detect excess alleles (derived alleles shared by P3 and P4) in P2, but not in a gene flow from P2 to P3.

This method can also infer the gene flow from their common ancestor (P12, P34). However, in our case, *P. mira* do not have a sister population. Thus, the recommended method is not applicable under this circumstances.

We then adopted $\phi_a\phi_i$ to estimate the expected gene flow between *P. mira* and the common ancestor of *P. kansuensis* and *P. persica*. The result showed that bidirectional gene flows were found between *P. mira* and the common ancestor of *P. kansuensis* and *P. persica*. However, the gene flow from this common ancestor to *P. mira* was significant stronger than the gene flow from *P. mira* to this common ancestor, supporting the result that we have proposed in the manuscript.

We could also notice that the divergence time of the involved species given by $\phi_a\phi_i$ was inconsistent with the result given by using BEAST approach. Considering that the divergence time from the BEAST approach meets well with the evidence of peach fossils 2.6 mya, is compatible with a series of topographical changes, and is consistent with the previously inferred divergence time of Peach and almond, we decided to emphasize this conclusion in the current study.

3. I am concerned about the choose of wild species used in the selective sweep analysis. The authors used all wild peaches (WP, line 252) including *P. mira*, *P. kansuensis*, *P. mongolica*, *P. tangutica*, and *P. davidiana* to identify selective signatures in the peach genome. This would complicate the identified signatures as some of them are not related to domestication but rather population differentiation, since some of the wild species are not involved in the domestication process at all. I understand that the direct wild progenitor of cultivated peaches remains unclear, but some species which are believed not to be the progenitor (e.g., *P. mira* based on this study?) should be excluded in the analysis to reduce the noises in the results.

Response:

Indeed, since the direct wild progenitor of cultivated peaches remains unclear, the extant peach wild relative species are unique resources available for studying the selection and domestication of peach. This is similar to a case study in soybean (*Glycine Max*) in which its wild relative species *Glycine soja* was used to study the impact of human selection on genetic variation in the soybean genome.

Here, according to this kind suggestion, we added new result by excluding *P. mira* from the selective sweep analysis (using both the SNP and CNV data from current study). We found that the *CNR* genes are remained in the sweep regions derived from these comparisons, supporting our former result that those genes likely play roles in increasing the fruit size during peach domestication. We also identified candidate genes that related to fruit texture, taste, *etc.*, and update our result in Supplementary Table 13–24. It has been reported that the fleshy mesocarp was the result of selection or domestication from the ancestor for the common ancestor of subg. *Amygdalus*¹, in which *P. mira*, *P. kansuensis*, and *P. persica* (*Persica* section species) developed fleshy and palatable mesocarps. Thus, we also retained the results based on all the of the wild relative peach species (*Persica* section in subg. *Amygdalus*) available in China for selective sweep analysis to avoid the loss of genomic background that existed in peach species.

Reference

1. Yazbek, M. & Oh, S. H. Peaches and almonds: phylogeny of *Prunus* subg. *Amygdalus* (Rosaceae) based on DNA sequences and morphology. *Plant Syst. Evol.* **299**, 1403–1418 (2013).

Minor concerns:

4. “peach (Rosaceae)” in the title is misleading. Rosaceae is a family that peach belongs to.

Response:

We have changed the word “Rosaceae” to “*Prunus persica*” in the title.

5. Line 21: change “Peach is” to "Peach (*Prunus persica* L. Batsch) is".

Response:

We have change “Peach” to "Peach (*Prunus persica*) is..." in the revised abstract. Now line 19.

6. Line 42, “estimated to 265 Mb”: should be “estimated haploid genome size of 265 Mb”.

Response:

We have change “estimated to 265 Mb” to “estimated haploid genome size of 265 Mb”. Now line 36.

7. Line 52-57: I couldn’t follow the logic here. How “indistinguishable endocarp” can suggest “edible fruits” (mesocarp)?

Response:

Thanks for this comment, we now appreciate that the sentence indeed had an ambiguity problem, and we have changed the expression in revised manuscript (Line 46–49), as:

“Surprisingly, peach endocarp fossils from 2.6-million-year-ago (Mya), found recently in Kunming, southwest China, are indistinguishable from endocarps of the modern peach cultivars, and studying of these fossils suggested that peaches may have acquired their modern-like edible fruits long prior to domestication, perhaps mediated by frugivorous primates⁷.”

8. Line 60-61: I don’t think fruit size and skin color are fruit edibility traits.

Response:

Yes, it is more appropriate to describe these as “fruit traits” in a category that includes fruit size, texture, taste, and skin color than to use “fruit edibility”. We have changed the expression consistently in our revised manuscript. Also see the description in revised manuscript (Line 57–58):

“Great morphological variation in fruit traits like fruit size, texture, taste, and skin color among cultivated peach and its wild relatives offers a natural diversity panel that presents an opportunity to explore the speciation and domestication history of peach...”

9. Line 78-84: Supplementary Figure 1 is not needed. It’s largely duplicated with Supplementary Table 2 and 4. Supplementary Table 3 should be removed. It’s not necessary. Just need to cite the genome paper or the link to the genome sequence. It’s very strange that in Supplementary Table 5 the numbers in the “Synonymous” column are almost all 15,138.

Response:

We have removed old Supplementary Figure 1 and old Supplementary Table 3, and have provided a link to the peach genome website in method section, and we have fixed the data filling problem in old Supplementary Table 5 (now Supplementary Table 4).

10. Line 100-103: need to cite the corresponding figure for this result.

Response:

We have added the citation for corresponding figure for this result. Now line 117.

11. Line 130, “among these five peach species”: Is almond (*P. dulcis*) a peach species?

Response:

Sorry for this obvious writing error; almond (*P. dulcis*) is not a peach species but rather belongs to *Amygdalus* section of subg. *Amygdalus*. We have changed the expression in revised manuscript (Line 153–156).

“It has been reported that self-fertilization in peach promotes the maintenance of extended LD¹⁸, so the increased extent of LD in the three peach species appears to have been impacted by the inbreeding in the evolutionary history of *Persica* section species.”

12. Line 158: change to “which is comparable to the size of modern peaches while remarkably larger than the fruit of extant non-edible peach species.”

Response:

We have changed “which is comparable to the size of modern peaches and is remarkably larger than the fruit of extant non-edible peach species” to “which is comparable to the size of modern peaches while remarkably larger than the fruit of extant non-edible peach species.” Now line 185.

13. Line 160-162: How the “0.60 Mya” was derived?

Response:

We have added the explanatory text to this sentence in the revised manuscript (now line 188-189) as follow:

“Based on an estimated time to the most recent common ancestor by estimating the divergence time between PL and PMC using all accessions in these two subgroups, we also inferred that the unknown direct wild ancestor of cultivated *P. persica* originated at least 0.60 Mya in the Pleistocene.”

14. Line 168 and 175: I suggest to mark the two periods (115-130 Kya and 19.0-26.5 Kya) on Figure 2b.

Response:

We have marked these two periods on Figure 2b with gray shadows.

15. Line 193: change “The early” to “Early”

Response:

We have changed “The early” to “Early”. Now line 222.

16. Line 194: change “this that” to “this”.

Response:

We have changed “this that” to “this”. Now line 223.

17. Line 348: change “Reads mapping” to “Read mapping”

Response:

We have changed “Reads mapping” to “Read mapping”. Now line 408.

18. Line 351: Provide a reference for SAMtools

Response:

The reference for SAMtools has been added to the reference list and indicated in the method. Now line 411.

19. Line 355: Change “SNPs annotation” to “SNP annotation”

Response:

We have changed “SNPs annotation” to “SNP annotation”. Now line 415.

20. Line 376: Change “reads ratio” to “ratio of reads”

Response:

We have changed “reads ratio” to “ratio of reads”. Now line 436.

21. Line 402-403: How the “substitution rate of 7.7×10^{-9} per site per generation and a generation time of 7 years” was derived?

Response:

A mutation rate $\mu = 7.7 \times 10^{-9}$ per site per generation was derived according to Xie *et al.* (2016), and the generation time of peach was derived from its full reproductive age of around 7 years according Wang *et al.*, 2012. and we mentioned it in the revised manuscript (now line 478-479).

Reference

1. Xie, Z. *et al.* Mutation rate analysis via parent–progeny sequencing of the perennial peach. I. A low rate in woody perennials and a higher mutagenicity in hybrids. *Proc. R. Soc. B.* **283**, 20161016 (2016).
2. Wang, L.R., Zhu, G.R. & Fang, W.C. Peach Genetic Resources in China. *China Agriculture Press*, (2012).

22. Line 428-429, “we assumed 1 cM was equal to 1 M bp”: this seems to be a wrong assumption. Can the authors provide data or references to support this assumption?

Response:

Thanks for your reminding of our inappropriate description here. It is not an assumption with biological significances. Actually, the ADMIXTOOLS took a default to 1Mb = 1 cM without an available linkage map in current study. Thus, we changed the description as: “The significance was assessed by a block jackknife procedure and a block size of 5 Mb was used.” in LINE 498-490. A reference involving a similar analysis was provided as below:

Reference

1. Fu Q, *et al.* An early modern human from Romania with a recent Neanderthal ancestor. *Nature* **524**, 216 (2015).

Reviewer #2 (Remarks to the Author):

The manuscript of Yu et al. (NCOMMS-18-05001) proposes a model for evolution of cultivated peach and related species from its origin in southwest China. The research combines genetic analysis of resequencing data from an ample collection of accessions including mume, almond, various peach wild relatives and cultivated varieties, with archaeological information, providing evidence for several highly relevant aspects of peach history and evolution, some of the most important being : 1) The establishment of the relationships between the group of *Prunus* species analyzed and the identification of introgression patterns between some of them; 2) the sequence of emergence of the different species, and their demographic story in relationship with the geological and climatic events of the different steps of evolution; 3) the evidence that fruit edibility was selected prior to domestication, possibly involving frugivores; 4) the indication that fruit size was selected earlier after

domestication and selection for fruit color is a much more recent event. Overall the paper provides a well-supported and singular evolutionary process for peach, different from that of other well studied perennial species such as apple and grape, and that may represent by itself a model for other tree or herbaceous crops. In addition, the paper provides useful information on candidate genes involved in the inheritance of key economical characters of interest for geneticists and breeders.

My view is that the manuscript describes excellent original research and is acceptable for publication in Nature Communications. I have only two comments and a few minor suggestions for improvement that are described below.

An interesting suggestion of the manuscript is that monkeys may have played a role in the selection for fruit edibility in peach and *Prunus mira* before domestication. As evidence for this, the Ne trajectories of three species during the Pleistocene are provided, two of which possibly sharing the same habitat with peach and another species that did not. The authors consider that the two former species shared similar trajectories but not the third, which is something that was not evident to me just by looking at Figure 2. A more detailed explanation or a statistical test supporting this trajectory similarity would help to make this stronger evidence.

Response:

Thanks for your enthusiasm and comments generally and for this kind suggestion in particular. We therefore added this result in the revised manuscript (now line 271-275) as:

“Subsequently, we used a Granger causality (GC) test³⁸ to assess if there is a similar trend in the effective population size (N_e) between the three monkey species and the peach species. Intriguingly, our GC results implied that the N_e of the two southwest monkey species (*Rhinopithecus brelichi* and *R. bieti*) can be used to predict the N_e trajectories of the three edible peach species (*P. mira*, *P. kansuensis*, and *P. persica*) (Supplementary Note 3).”

The authors identify a number of putative selection sweeps comparing the wild, landrace and modern cultivar compartments, and then find in these regions genes involved in fruit characteristics of agricultural interest. Some relevant examples are provided suggesting selection for genes known to affect these characters in other species that are located in the selection sweeps. This is a reasonable approach, but one could argue that provided a sufficiently large number of candidate genes, the probability of finding one or several of these genes by chance in the genome fragments identified is high. Would it be possible for one or more of these characters to estimate the total number of genes possibly involved and to test whether the numbers of genes identified are significantly higher than those expected by chance?

Response:

Thanks for your comment. Working from the assumption that the selected fruit-related genes are enriched in the selective sweep regions, we estimated the putative “threshold number” for fruit-related genes in the peach genome: using Fisher's exact test (significant P -value < 0.05), one would expect no more than 1,296 (WP vs. CP), 1,626 (WP vs. PL), or 576 (PL vs. PMC) fruit-related genes should present on peach genome. However, given that there is no designated ‘fruit-related’ gene list, it is hard to estimate the exact significance of the putative selected fruit genes number in sweep regions. Note that we chose candidate genes from among the top 1% of selective sweep regions, so there was a strict and reliable selection cutoff in our study of the domestication of fruit traits. Also, consider that a limited number of key genes may play hugely influential roles in driving a “domestication syndrome” during crop domestication; under this circumstance, we would not expect a significantly higher number of genes in a candidate sweep region.

Minor changes proposed:

Abstract

Line 23: change “... species are closely...” to species “... species is closely...”

Response:

We have changed “... species are closely...” to species “... species is closely...”. Now line 70 in the main text.

Line 31: change “...during peach...” to “...during modern peach...”

Response:

We have changed “...during peach...” to “...during modern peach...”. Now line 76 in the main text.

Introduction

Line 72. "...our novel hypothesis...". This hypothesis was already proposed in the paper of Su et al. (2015), cited in the manuscript.

Response:

We have modified the expression in this sentence and cited the paper. Now line 71-73.

Results

Line 97. Change "...maturity and morphologically..." to "...maturity and are morphologically..."

Response:

We have changed "...maturity and morphologically..." to "...maturity and are morphologically...".
Now line 112.

Line 194. Change "...and this that..." to "...and this..."

Response:

We have changed "...and this that..." to "...and this...". Now line 223.

Line 233. Change "...the basis characteristics..." to "...the basic characteristics ..."

Response:

We have changed "...the basis characteristics..." to "...the basic characteristics ...". Now line 262.

Lines 283-284. A reference could be provided to support the statement that stone and fruit size are strongly correlated.

Response:

The correlation between the size of a peach stone and fruit size has been reported in Zheng *et al.*, 2014, and we added the citation to this conclusion in the revised manuscript. Now line 322.

Methods

Line 421. Change "...based genome-wide..." to "...based on genome-wide..."

Response:

We have changed "...based genome-wide..." to "...based on genome-wide...". Now line 483.

Figure Legends

Line 676. Change "...Linage..." to "...Linkage..."

Response:

We have changed "...Linage..." to "...Linkage...".

Supplementary Figure 6. The *P. ferganensis* group appears to be the one marked with dark green color, but it is not sufficiently clear. It could be stated on the legend or indicated on the diagram.

Response:

We have modified the old Supplementary Figure 6 (now the Supplementary Figure 5) and added the statement to the legend following the reviewer's helpful suggestion.

Reviewer #3 (Remarks to the Author):

The authors used 58 high-coverage genomes of cultivated peach and closely related relatives to explore the evolutionary history, including the genetic divergence of cultivated peaches and its closely related relatives, demographic trajectories, domestication features and their evolved time. The topics addressed are interesting, but there are several problems with the manuscript as presented.

The exploration of data analysis with appropriate tools is quite weak and some of the conclusions are lack of evidences to support. There are several issues with the manuscript that need to be addressed before the manuscript is published.

Major issues:

1. SNP calling and filtering is not clear enough. The author reported that the total SNPs called are 24,280,369, but it is unclear why only use 2,559,589 SNPs to do PCA and 3,909,617 to build NJ tree and how they did filtering. Please also clarify how the author did SNP validation with SNP array data and how many of those SNPs can be found in the resequencing data. In order to see if the SNPs called from whole genome resequencing data are reasonable, please show the site frequency spectrum (SFS) and the ratio of transition and transversion (Ts:Tv). Since the author did annotations for all of the SNPs with Annovar, but it is unclear how many of those SNPs are noncoding, coding, synonymous or nonsynonymous SNPs.

Response:

Thanks for the comment. We generated two sets of results: **1)** firstly using all 58 accessions and *P. mume* (Supplementary Fig. 2), and **2)** then removed the non-*Amygdalus* accessions for the 51 *Amygdalus* accessions (Fig. 1b,c). The total SNPs were called based on all 58 accessions which included the species that showed large genetic differences compared to the reference genome of peach, leading to a total of 24,280,369 SNPs. The PCA and NJ tree in Fig. 1 was built based on the SNPs from 51 *Amygdalus* accessions, which have been filtered as follows: PCA: maf < 5%, call rate > 90%; NJ-tree: maf < 5%, call rate > 90% (also mentioned in method).

We have developed an Axiom® Peach 170K SNP array with SNPs discovered in 96 peach accessions (including the accessions in this study, and from two previous studies, Verde *et al.*, 2013 and Cao *et al.*, 2014). Since this SNP array was developed based on the *P. persica* Whole Genome v1.0 reference genome, we thus applied the liftover program to transit the genotyped SNPs from *P. persica* Whole Genome v1.0 reference genome to *P. persica* Whole Genome v2.1.a1 for SNP validation as we described in the manuscript.

We have added the analysis of SFS (distribution of MAF) (Supplementary Fig. 8) and addressed this result in the revised manuscript. The number of Ts and Tv and the value of Ts:Tv for

each accession were calculated and are provided in Supplementary Table 4. All SNP annotation information for each accession is also presented in Supplementary Table 4; this contains information including noncoding, coding, synonymous or nonsynonymous SNPs, the Ts/Tv ratio, *etc.*

2. Some of the analyses applied are either potentially inappropriate or over interpreted. For example, the authors make use of ADMIXTURE for population structure. However, ADMIXTURE likely makes a Hardy Weinberg assumption regarding expected genotypes. Violations of this assumption could lead to over clustering of samples owing to inbreeding. Did the authors have a strategy for addressing this in samples that are highly inbred?

Response:

Thanks for this comment. ADMIXTURE assumes that loci are at HWE within populations, and many programs that use allele frequencies to analyze genetic data assume HWE at some point, and it is now appreciated that HWE is seldom met in practical studies. Thus, we indeed need to cautiously interpret our results by combining more additional analysis. We therefore also included PCA, NJ-tree, IS, IBS, and MJ-networking analysis to manage any spurious findings that could have come from any one particular method with the goal of more comprehensively addressing the population structure of these species.

In order to reduce this affect, we filtered SNPs by testing the HWE violation ($P > 10^{-4}$) and reconstructed the model-based clustering result as follow:

This model-based clustering result ($K = 9$) also supported the grouping pattern that given by NJ tree, and we could also draw the same conclusion that *P. ferganensis* is a geographical ecotype of *P.*

P. persica, as the *P. ferganensis* accessions were clustered with the *P. persica* landraces from north China ($K = 7$).

In summary, we believed that the ADMIXTURE tool, as has been widely used in population structure study of many domesticated crops including species with extreme high inbreeding levels like rice (a heterozygosity of 0.0038%)¹, does provide valuable information to investigate the population structure. However, we agree and have emphasized in the writing of our revised manuscript that cautious interpretation should be addressed by combining additional analyses and more biological background.

Reference

1. Du, H. *et al.* Sequencing and de novo assembly of a near complete indica rice genome. *Nat. Commun.* **8**, 15324 (2017).

3. The authors used VCFtools to calculate genetic nucleotide diversity, Tajima's D and Fst. VCFtools were designed to calculate the population genetic statistics for diploid samples, but the majority of the samples in this study are highly selfing (which will be treated as haploid), and can not VCFtools to calculate genetic nucleotide diversity, Tajima's D and Fst.

Response:

Kindly note that the other major model species for fruit genomics research—apple and grape—are self-incompatible, so we were guided in our analysis by studies in the self-compatible model plant rice. Although *P. persica* is self-compatible, it shows a higher heterozygosity (0.126%) than human (a heterozygosity of 0.43×10^{-3})^{1,2}. While, in studies of human, samples are not typically treated as haploid for the calculation of nucleotide diversity, Tajima's D, or *Fst* values.

References:

1. Wang, J. *et al.* The diploid genome sequence of an Asian individual. *Nature* **456**, 60–65 (2008).

2. Li, R. *et al.* SNP detection for massively parallel whole-genome resequencing. *Genome Res.* **19**, 1124–1132 (2009).

4. All of the scripts and analytical tools used for a largely computational project like the work reported here need to be made available through Github, Bitbucket, or a similar version control repository. Because the scripts are essential to reproducing this work, they should be available to reviewers and/or readers. These resources are important in part because samtools, GATK, and similar utilities include a large number of options that alter the resulting output.

Response:

According to your kind advice, we have uploaded the primary code used in the current study to Github:

https://github.com/ColyFu/peach_edibility/blob/master/work.sh

5. Some of the arguments or conclusions seem to be lack of the evidences to support. For example, in Line 129, the authors argued that "it appears that inbreeding level has likely played a major role in determining the level of LD among these five peach species", but apparently the authors did not examine the influences of other factors except inbreeding. Another example is in line 237 to line 245, the authors proposed that " the ancient selection process likely involved frugivorous, contributing to the emergence of fruit edibility, prior to recent human-mediated domestication" only based on that the N_e trajectories of three monkey species has the similar pattern with the peaches in this study.

Response:

Thanks for your comments. Yes, Linkage disequilibrium is influenced by many factors, including selection, rate of recombination, mutation, genetic drift, mating system, population structure, and genetic linkage, and it has been reported that mating system differences between closely related species pairs can significantly affect many aspects of genome evolution in *Arabidopsis*, *Capsella*, and *Collinsia* (including lower nucleotide diversity, higher linkage disequilibrium (LD), and reduced effective population size (N_e))¹. For *Prunus specis*, it is believed that SI is predominant although self-compatibility (SC) forms occur in *P. persica*. According to our results, the increasing in the inbreeding level in *P. mira*, *P. kansuensis*, and *P. persica* (as evidenced by ROH lengths and F values) likely contribute to the high extent of LD in these species. This conclusion also mirrors a previous study² which proposed that self-fertilization in peach favors the maintenance of extended LD. It is of course complicated to fully interpret the evolutionary consequences of the relationship between selection/domestication, inbreeding, and LD, thus, we have change the expression in the revised manuscript (now line 134-156).

References

1. Velasco, D. *et al.* Evolutionary genomics of peach and almond domestication. *G3: Genes, Genomes, Genetics* **6**, 3985-3993 (2016).
2. Cao, K. *et al.* Genome-wide association study of 12 agronomic traits in peach. *Nat. Commun.* **7**, 13246 (2016).

Thanks for your comments. We thus performed Granger causality test for assessing N_e trend similarity between monkey and peach species to provide more statistical support for our proposal that the pre-historical animal-plant interaction for peach is plausible in the revised manuscript (now line 271-275) as:

“Subsequently, we used a Granger causality (GC) test³⁸ to assess if there is a similar trend in the effective population size (N_e) between the three monkey species and the peach species. Intriguingly, our GC results implied that the N_e of the two southwest monkey species

(*Rhinopithecus brelichii* and *R. bieti*) can be used to predict the N_e trajectories of the three edible peach species (*P. mira*, *P. kansuensis*, and *P. persica*) (Supplementary Note 3).”

Previous studies have proposed that the ancestry of the common ancestor of *Amygdalus* sub species had dry, split, inedible fruit, and was most self-incompatible, while the transition to fleshy edible mesocarp seems to have resulted from the selection/domestication traits in peach species^{1,2}. We also found “genomic footprints” that likely result from positive selection in the three edible peach species. We added some new context in revised manuscript in line 126-129 and line 143-149.

Additionally, the study of peach fossils evidence implies that frugivores likely selected peaches prior to a later human mediated domestication³. As a matter of fact, it is commonly found in zoecology studies that frugivores (also known as dispersers) can lead dispersal syndromes for fruit traits^{4,5}. We thus discuss the idea that frugivores may have influenced the aforementioned pre-history selection by contributing to the emergence of fleshy fruit in peach species in the revised manuscript, and we also added some viewpoints from previous studies for a fuller discussion of the evolution of fleshy mesocarp in peach species, hopefully thereby adding valuable information about this issue. See more additional details in discussion section in revised manuscript.

Reference

1. Yazbek, M. & Oh, S. H. Peaches and almonds: phylogeny of *Prunus* subg. *Amygdalus* (Rosaceae) based on DNA sequences and morphology. *Plant Syst. Evol.* **299**, 1403-1418 (2013).
2. Yazbek, M. & Al-Zein, M. S. Wild almonds gone wild: revisiting Darwin’s statement on the origin of peaches. *Genet Resour Crop Evol.*, **61**, 1319-1328 (2014).
3. Su, T., Wilf, P., Huang, Y., Zhang, S. & Zhou, Z. Peaches preceded humans: fossil evidence from SW China. *Sci. Rep.* **5**, srep16794 (2015).
4. Lomáscolo, S. B. *et al.* Correlated evolution of fig size and color supports the dispersal syndromes hypothesis. *Oecologia* **156**, 783-796 (2008).
5. Valenta, K. *et al.* Colour and odour drive fruit selection and seed dispersal by mouse lemurs. *Sci. Rep.* **3** 2424 (2013).

6. The author argued that " it is reasonable to infer that common ancestor of *P. persica* and *P. kansuensis* was self-incompatible. We therefore propose that the combination-incompatibility and overlapping habits...approximately 3 million years ago". Is *P.kansuensis* self-incompatible? Why is it reasonable to infer that common ancestor of *P.persica* and *P.kansuensis* was self-incompatible? Again it is hard to follow the statement that "We therefore propose that the

combination-incompatibility and overlapping habits...approximately 3 million years ago". How did the author get this conclusion?

Response

Thanks for your comments, we have modified our description in the new manuscript (line 242-249) as follow:

“*P. persica* has a high level of inbreeding owing to its self-compatibility. It has been proposed that self-incompatibility (SI) is predominant in *Prunus*⁹, and considering that the transition to self-compatibility (SC) is known to have been caused by a loss-of-function mutation³², we infer that populations of the common ancestor of *P. kansuensis* and *P. persica* likely contained some SI individuals that would have enabled cross-pollination. We therefore propose that the combination of a capacity for cross-pollination and overlapping habitats (before the outwards dispersal and domestication of *P. persica*) created a highly favorable context for the ancient introgressions between *P. mira* and the common ancestor of *P. persica* and *P. kansuensis* approximately 3 million years ago.”

7. Line 221-223, how did the author know stronger signatures in introgressed segments compared with genome levels of *P. mira*? Based on which comparison?

Response:

We have added the description to the revised manuscript. The comparison was performed using a bi-population selective sweep method, XP-EHH, and the *P*-value is presented in the main text (line 250-252).

Minor issues:

1. Line47, please fix the citation format.

Response:

We modified this citation format and added the reference (now line 41).

2. Line 114, "An increase in the ratio of nonsynonymous to synonymous SNPs.." in line 114, did the author use the private SNPs to do calculation?

Response:

No. We used all of the SNPs for the analysis of the ratio of nonsynonymous to synonymous SNPs.

3. Which tools did the author to calculate the ROH? Please clarify in the Method.

Response:

We used PLINK for the calculation, and we have provided the information about the calculation of ROHs in the methods section (line 443-449).

4. Which tool did the authors calculate the inbreeding coefficient? Please clarify in the Method.

Response:

The coefficient of inbreeding (F value) in each population was calculated using the formula $F = 1 - H_o/H_e$, and we have clarified the calculation of inbreeding coefficient in method (line 450-452).

5. Please cite the paper for the substitution rate of 7.7×10^{-9} in the main text.

Response:

We have added the description of the substitution rate (7.7×10^{-9}) in the main text (now line 159).

6. Please label the samples consistently. For example, use those labels CA, WM, WD, WK, CP in all of the figures, tables, and supplemental tables, otherwise it is hard to follow.

Response:

We re-labeled the samples consistently by changing their captions appropriately with WM, WD, or WK, while for CP (cultivated peach, *P. persica*) and CA (cultivated almond, *P. dulcis*), both of which included modern cultivar and landrace accessions, we refer to them as: PL (*Persica* landrace) and PMC (*Persica* modern cultivar), and DL (*Dulcis* landrace), and DMC (*Dulcis* modern cultivar) in the revised manuscript.

Reviewer #1 (Remarks to the Author):

The authors addressed all my concerns. Here are some minor changes I would suggest:

Line 19: change “for study” to “for studying”

Line 20: Change “We here explored” to “Here we explore”

Line 29: “an indispensable instance” seems not accurate. May change to “valuable information”.

Line 41: change “with annual” to “with an annual”

Line 68: Reference 9 seems wrongly cited here. Please check the reference citation across the manuscript.

Line 90: remove “with GATK”.

Line 93: change “principle” to “principal”

Line 133: Remove “data”

Line 181: Change “accurate” to “the case”

Line 196: Change “began” to “starting”

Line 223 and 252: add references for TreeMix and XP-EHH

Line 262: Change “basis” to “basic”

Line 288: remove “and this especially so in peach”

Reviewer #2 (Remarks to the Author):

The authors have addressed correctly my comments from the first review. I think that the current versión is acceptable for publication in Nature Communications.

I have found a few minor errors that should be corrected by the authors:

Line 93 and legend of Supp Fig 2: change principle to principal

Line 138 delete "Rosaceae"; grapewine is nor a rosaceous crop

Line 384 change "process" to "processes"

Reviewer #3 (Remarks to the Author):

The author solved majority of the my comments, but some of the explanation are not reflected in the main text. Also a couple of my comments are not well explained.

Unsolved major issues:

1. SNP calling and filtering is not clear enough. The author reported that the total SNPs called are 24,280,369, but it is unclear why only use 2,559,589 SNPs to do PCA and 3,909,617 to build NJ tree and how they did filtering. Please also clarify how the author did SNP validation with SNP array data and how many of those SNPs can be found in the resequencing data. In order to see if the SNPs called from whole genome resequencing data are reasonable, please show the site frequency

spectrum (SFS) and the ratio of transition and transversion (Ts:Tv). Since the author did annotations for all of the SNPs with Annovar, but it is unclear how many of those SNPs are noncoding, coding, synonymous or nonsynonymous SNPs.

Response:

Thanks for the comment. We generated two sets of results: 1) firstly using all 58 accessions and *P. mume* (Supplementary Fig. 2), and 2) then removed the non-*Amygdalus* accessions for the 51 *Amygdalus* accessions (Fig. 1b,c).

Please reflect this in the caption of Fig. 1b

The total SNPs were called based on all 58 accessions which included the species that showed large genetic differences compared to the reference genome of peach, leading to a total of 24,280,369 SNPs. The PCA and NJ tree in Fig. 1 was built based on the SNPs from 51 *Amygdalus* accessions, which have been filtered as follows: PCA: maf < 5%, call rate > 90%; NJ-tree: maf < 5%, call rate > 90% (also mentioned in method).

Please clarify the filtering criteria in the caption of Fig. 1b.

We have developed an Axiom® Peach 170K SNP array with SNPs discovered in 96 peach accessions (including the accessions in this study, and from two previous studies, Verde et al., 2013 and Cao et al., 2014). Since this SNP array was developed based on the *P. persica* Whole Genome v1.0 reference genome, we thus applied the liftover program to transit the genotyped SNPs from *P. persica* Whole Genome v1.0 reference genome to *P. persica* Whole Genome v2.1.a1 for SNP validation as we described in the manuscript.

Please add this in the method part of the main text or at it in the supplemental notes.

We have added the analysis of SFS (distribution of MAF) (Supplementary Fig. 8) and addressed this result in the revised manuscript. The number of Ts and Tv and the value of Ts:Tv for each accession were calculated and are provided in Supplementary Table 4. All SNP annotation information for each accession is also presented in Supplementary Table 4; this contains information including noncoding, coding, synonymous or nonsynonymous SNPs, the Ts/Tv ratio, etc.

2. Some of the analyses applied are either potentially inappropriate or over interpreted. For example, the authors make use of ADMIXTURE for population structure. However, ADMIXTURE likely makes a Hardy Weinberg assumption regarding expected genotypes. Violations of this assumption could lead to over clustering of samples owing to inbreeding. Did the authors have a strategy for addressing this in samples that are highly inbred?

Response:

Thanks for this comment. ADMIXTURE assumes that loci are at HWE within populations, and many programs that use allele frequencies to analyze genetic data assume HWE at some point, and it is now appreciated that HWE is seldom met in practical studies. Thus, we indeed need to cautiously interpret our results by combining more additional analysis. We therefore also included PCA, NJ-tree, IS, IBS, and MJ-networking analysis to manage any spurious findings that could have come from any one particular method with the goal of more comprehensively addressing the population structure of these species.

In order to reduce this affect, we filtered SNPs by testing the HWE violation ($P > 10^{-4}$) and reconstructed the model-based clustering result as follow:

Please add this part in the Method of the main text and update the Supplementary fig5

This model-based clustering result ($K = 9$) also supported the grouping pattern that given by NJ tree, and we could also draw the same conclusion that *P. ferganensis* is a geographical ecotype of *P. persica*, as the *P. ferganensis* accessions were clustered with the *P. persica* landraces from north China ($K = 7$).

In summary, we believed that the ADMIXTURE tool, as has been widely used in population structure study of many domesticated crops including species with extreme high inbreeding levels like rice (a heterozygosity of 0.0038%)¹, does provide valuable information to investigate the population structure. However, we agree and have emphasized in the writing of our revised manuscript that cautious interpretation should be addressed by combining additional analyses and more biological background.

I did not see you emphasized in the revised version. Please reflect in the main text and point it out in the Rebuttal letter.

Reference

1. Du, H. et al. Sequencing and de novo assembly of a near complete indica rice genome. *Nat. Commun.* 8, 15324 (2017).

6. The author argued that " it is reasonable to infer that common ancestor of *P. persica* and *P. kansuensis* was self-incompatible. We therefore propose that the combination-incompatibility and overlapping habits...approximately 3 million years ago". Is *P.kansuensis* self-incompatible? Why is it reasonable to infer that common ancestor of *P.persica* and *P.kansuensis* was self-incompatible? Again it is hard to follow the statement that "We therefore propose that the combination-incompatibility and overlapping habits...approximately 3 million years ago". How did the author get this conclusion?

Response

Thanks for your comments, we have modified our description in the new manuscript (line 242-249) as follow:

"*P. persica* has a high level of inbreeding owing to its self-compatibility. It has been proposed that self-incompatibility (SI) is predominant in *Prunus*⁹, and considering that the transition to self-compatibility (SC) is known to have been caused by a loss-of-function mutation³², we infer that populations of the common ancestor of *P. kansuensis* and *P. persica* likely contained some SI individuals that would have enabled cross-pollination. We therefore propose that the combination of a capacity for cross-pollination and overlapping habitats (before the outwards dispersal and domestication of *P. persica*) created a highly favorable context for the ancient introgressions between *P. mira* and the common ancestor of *P. persica* and *P. kansuensis* approximately 3 million years ago."

The author did not answer my question " Is *P.kansuensis* self-incompatible?". Also does the SI locus in the introgression region? What does "overlapping habitats" mean? Please clarify.

Response to Reviewers' comments

Reviewers' comments:

Reviewer #1 (Remarks to the Author):

The authors addressed all my concerns. Here are some minor changes I would suggest:

Line 19: change “for study” to “for studying”

Response:

Thanks for your continued input about our study; we appreciate your help and guidance. We have changed “for study” to “for studying” in the revised abstract. Now line 19.

Line 20: Change “We here explored” to “Here we explore”

Response:

We have changed “We here explored” to “Here we explore” in the revised abstract. Now line 20.

Line 29: “an indispensable instance” seems not accurate. May change to “valuable information”.

Response:

We have changed “an indispensable instance” to “valuable information” in the revised abstract. Now line 28-29.

Line 41: changed “with annual” to “with an annual”

Response:

We have changed “with annual” to “with an annual” in the revised manuscript. Now line 40-41.

Line 68: Reference 9 seems wrongly cited here. Please check the reference citation across the manuscript.

Response:

Thanks for your bringing this to our attention. We have re-checked reference 9: “Wild almonds gone wild: revisiting Darwin’s statement on the origin of peaches”, in which the author wrote that “Alternatively, the fleshy mesocarp characterizing the peach subclade may be a byproduct of domestication. Many studies suggest that *P. mira*, *P. ferganensis* and *P. kansuensis* are feral types of the cultivated peach, supporting the hypothesis that the loss of a dry splitting mesocarp is the result of artificial selection for fleshiness during domestication”.

Line 90: remove “with GATK”.

Response:

We have removed “with GATK” in the revised manuscript.

Line 93: change “principle” to “principal”

Response:

We have changed “principle” to “principal” in the revised manuscript. Now line 92.

Line 133: Remove “data”

Response:

We have removed “data” in the revised manuscript.

Line 181: Change “accurate” to “the case”

Response:

We have changed “accurate” to “the case” in the revised manuscript. Now line 183.

Line 196: Change “began” to “starting”

Response:

We have changed “began” to “starting” in the revised manuscript. Now line 198.

Line 223 and 252: add references for TreeMix and XP-EHH

Response:

We added the references for TreeMix and XP-EHH in the revised manuscript. Now line 224 and 246, respectively.

Line 262: Change “basis” to “basic”

Response:

We have changed “basis” to “basic” in the revised manuscript. Now line 256.

Line 288: remove “and this especially so in peach”

Response:

We have removed “and this especially in peach” in the revised manuscript. Now line 281.

Thanks again for your work on our behalf.

Reviewer #2 (Remarks to the Author):

The authors have addressed correctly my comments from the first review. I think that the current versión is aceptable for publication in Nature Communications.

I have found a few minor errors that should be corrected by the authors:

Line 93 and legend of Supp Fig 2: change principle to principal

Response:

We appreciate your ongoing support and are thankful for your continued guidance for our study.

We have changed “principle” to “principal” in the revised manuscript. Now line 92 and legend of Supplementary Fig. 2.

Line 138 delete "Rosaceae"; grapewine is nor a rosaceous crop

Response:

Sorry for this obvious error. We have deleted "Rosaceae" in the revised manuscript. Now line 142.

Line 384 change "process" to "processes"

Response:

We have changed "process" to "processes" in the revised manuscript. Now line 376.

Thanks for your continued help in improving our study.

Reviewer #3 (Remarks to the Author):

The author solved majority of the my comments, but some of the explanation are not reflected in the main text. Also a couple of my comments are not well explained.

Unsolved major issues:

1. SNP calling and filtering is not clear enough. The author reported that the total SNPs called are 24,280,369, but it is unclear why only use 2,559,589 SNPs to do PCA and 3,909,617 to build NJ tree and how they did filtering. Please also clarify how the author did SNP validation with SNP array data and how many of those SNPs can be found in the resequencing data. In order to see if the SNPs called from whole genome resequencing data are reasonable, please show the site frequency spectrum (SFS) and the ratio of transition and transversion (Ts:Tv). Since the author did annotations for all of the SNPs with Annovar, but it is unclear how many of those SNPs are noncoding, coding, synonymous or nonsynonymous SNPs.

Original Response: "Thanks for the comment. We generated two sets of results: 1) firstly using all 58 accessions and P. mume (Supplementary Fig. 2), and 2) then removed the non-Amygdalus accessions for the 51 Amygdalus accessions (Fig. 1b,c)."

Please reflect this in the caption of Fig. 1b

Response:

Thanks for your continued guidance to improve our manuscript. As for this suggestion. We now present these details in the caption for Fig. 1b.

Original Response: "The total SNPs were called based on all 58 accessions which included the species that showed large genetic differences compared to the reference genome of peach, leading to a total of 24,280,369 SNPs. The PCA and NJ tree in Fig. 1 was built based on the SNPs from 51 Amygdalus accessions, which have been filtered as follows: PCA: maf < 5%, call rate > 90%; NJ-tree: maf < 5%, call rate > 90% (also mentioned in method)."

Please clarify the filtering criteria in the caption of Fig. 1b.

Response:

Thanks for this suggestion. We have clarified the filtering criteria in the caption of Fig. 1b.

Original Response: "We have developed an Axiom® Peach 170K SNP array with SNPs discovered in 96 peach accessions (including the accessions in this study, and from two previous studies, Verde et al., 2013 and Cao et al., 2014). Since this SNP array was developed based on the P. persica Whole Genome v1.0 reference genome, we thus applied the liftover program to transit the genotyped SNPs from P. persica Whole Genome v1.0 reference genome to P. persica Whole Genome v2.1.a1 for SNP validation as we described in the manuscript."

Please add this in the method part of the main text or at it in the supplemental notes.

Response:

We have added this SNP validation details in the Methods section of the main text. Now line 406-412.

Original Response: “We have added the analysis of SFS (distribution of MAF) (Supplementary Fig. 8) and addressed this result in the revised manuscript. The number of Ts and Tv and the value of Ts:Tv for each accession were calculated and are provided in Supplementary Table 4. All SNP annotation information for each accession is also presented in Supplementary Table 4; this contains information including noncoding, coding, synonymous or nonsynonymous SNPs, the Ts/Tv ratio, etc.”

2. Some of the analyses applied are either potentially inappropriate or over interpreted. For example, the authors make use of ADMIXTURE for population structure. However, ADMIXTURE likely makes a Hardy Weinberg assumption regarding expected genotypes. Violations of this assumption could lead to over clustering of samples owing to inbreeding. Did the authors have a strategy for addressing this in samples that are highly inbred?

Original Response: “Thanks for this comment. ADMIXTURE assumes that loci are at HWE within populations, and many programs that use allele frequencies to analyze genetic data assume HWE at some point, and it is now appreciated that HWE is seldom met in practical studies. Thus, we indeed need to cautiously interpret our results by combining more additional analysis. We therefore also included PCA, NJ-tree, IS, IBS, and MJ-networking analysis to manage any spurious findings that could have come from any one particular method with the goal of more comprehensively addressing the population structure of these species.”

Original Response: “In order to reduce this affect, we filtered SNPs by testing the HWE violation ($P > 10^{-4}$) and reconstructed the model-based clustering result as follow:...”

Please add this part in the Method of the main text and update the Supplementary fig5

Response:

We have added the model-based clustering analysis that used filtered SNPs and tested HWE violations in the Methods section of the main text (Now line 425-427); we have also updated Supplementary Fig. 5 according to your helpful suggestion.

*Original Response: “This model-based clustering result ($K = 9$) also supported the grouping pattern that given by NJ tree, and we could also draw the same conclusion that *P. ferganensis* is a geographical ecotype of *P. persica*, as the *P. ferganensis* accessions were clustered with the *P. persica* landraces from north China ($K = 7$).”*

Original Response: “In summary, we believed that the ADMIXTURE tool, as has been widely used in population structure study of many domesticated crops including species with extreme high inbreeding levels like rice (a heterozygosity of 0.0038%)1, does provide valuable information to investigate the population structure. however, we agree and have emphasized in the writing of our revised manuscript that cautious interpretation should be addressed by combining additional analyses and more biological background.”

I did not see you emphasized in the revised version. Please reflect in the main text and point it out in the Rebuttal letter.

Response:

Thanks for your helpful suggestion. Considering the potential HWE violation, we have added the result of model-based clustering analysis using filtered SNPs by testing the HWE violation and have updated the main text of the paper to more comprehensively address this important concern (Now line 120-125). The updated text for this point from the main manuscript reads as follows:

“ADMIXTURE, which is widely used and provides valuable information for investigating the population structures of many domesticated crops—including species with extreme high inbreeding level like rice¹⁶—assumes that loci are at HWE (Hardy-Weinberg equilibrium) within populations. Considering the potential impact of HWE violations, we further filtered SNPs by testing HWE violation ($P > 10^{-4}$) and performed model-based clustering analysis; this analysis gave to a consistent conclusion about the population structure ($K = 9$; Supplementary Fig. 5b).”

6. The author argued that " it is reasonable to infer that common ancestor of *P. persica* and *P. kansuensis* was self-incompatible. We therefore propose that the combination-incompatibility and overlapping habits...approximately 3 million years ago". Is *P.kansuensis* self-incompatible? Why is it reasonable to infer that common ancestor of *P.persica* and *P.kansuensis* was self-incompatible? Again it is hard to follow the statement that "We therefore propose that the combination-incompatibility and overlapping habits...approximately 3 million years ago". How did the author get this conclusion?

Original Response: "Thanks for your comments, we have modified our description in the new manuscript (line 242-249) as follow:

"P. persica has a high level of inbreeding owing to its self-compatibility. It has been proposed that self-incompatibility (SI) is predominant in Prunus⁹, and considering that the transition to self-compatibility (SC) is known to have been caused by a loss-of-function mutation³², we infer that populations of the common ancestor of P. kansuensis and P. persica likely contained some SI individuals that would have enabled cross-pollination. We therefore propose that the combination of a capacity for cross-pollination and overlapping habitats (before the outwards dispersal and domestication of P. persica) created a highly favorable context for the ancient introgressions between P. mira and the common ancestor of P. persica and P. kansuensis approximately 3 million years ago."

The author did not answer my question " Is *P.kansuensis* self-incompatible?". Also does the SI locus in the introgression region? What does "overlapping habitats" mean? Please clarify.

Response:

Thank you for focusing our attention on this aspect of our manuscript. In the course of our revision process, we eventually came to the conclusion that the best course of action was to remove these speculative claims from our manuscript; we have done this. Here, seeking to address

your questions about our initially included speculative claim, we simply revisit the data which had motivated our now-removed conclusions, and present the answers as follow:

The author did not answer my question " Is *P.kansuensis* self-incompatible?"

Response:

Specifically, whereas *P. kansuensis* is regard as a self-compatible species^{1,2}, we found that both *P. mira* and *P. kansuensis* showed lower inbreeding level than *P. persica*, but higher inbreeding level than *P. dividiana* and *P. dulcis* (Fig. 1d).

We initially interpreted this to indicate that perhaps relatively more cross-pollination had occurred in these two peach species. We then surmised that, given both the proposal that self-incompatibility (SI) is predominant in *Prunus* and the evidence of a gradual transition to self-compatibility (SC) in some *Prunus* section peach species, that the common ancestor of *P. kansuensis* and *P. persica* could have had a lower inbreeding level than *P. persica*. This would have, in theory, enabled more cross-pollination and would have potentially encouraged the introgression between *P. mira* and the common ancestor of *P. kansuensis* and *P. persica*.

References:

1. Zheng, Y., Crawford, G. & Chen, X. Archaeological evidence for peach (*Prunus persica*) cultivation and domestication in China. *PLoS One*, e106595 (2014).
2. Velasco, D., Hough, J., Aradhya, M. & Rossibarra, J. Evolutionary genomics of peach and almond domestication. *G3: Genes/Genomes/Genetics*. 6, 3985-3993 (2016).

Also does the SI locus in the introgression region?

Response:

We scanned the introgressed segments, and found that the S-locus is not involved in this introgression.

What does "overlapping habitats" mean?

Response:

The term that we had used but have now removed from our manuscript ("overlapping habitats")_was intended to refer to an coinciding area in which both the *P. mira* and *P. kansuensis* habitats were located (northeast of the Tibetan plateau, south of current Gansu province, and west of current Sichuan province) (Fig.1a). Hence, we had proposed that “the combination of a capacity for cross-pollination and overlapping habitats (before the outwards dispersal and domestication of *P. persica*) created a highly favorable context for the ancient introgressions between *P. mira* and the common ancestor of *P. persica* and *P. kansuensis*”.

Guided by your concerns, we now appreciate that this speculative conclusion was not strong enough to merit inclusion in our manuscript. Clearly, further studies will be needed to more comprehensively address SI or SC in peach species.

We again thank the reviewer for their guidance and care in helping us to improve our manuscript.

** See Nature Research's author and referees' website at www.nature.com/authors for information about policies, services and author benefits

This email has been sent through the Springer Nature Tracking System NY-610A-NPG&MTS

Confidentiality Statement:

This e-mail is confidential and subject to copyright. Any unauthorised use or disclosure of its contents is prohibited. If you have received this email in error please notify our Manuscript Tracking System Helpdesk team at <http://platformsupport.nature.com>.

Details of the confidentiality and pre-publicity policy may be found here <http://www.nature.com/authors/policies/confidentiality.html>

Privacy Policy | Update Profile

Reviewer #3 (Remarks to the Author):

The authors have addressed correctly my comments from the second review. I think that the current version is acceptable for publication in Nature Communications.

Supplementary peer review

Response to Reviewers' comments

The first review

Reviewer #1 (Remarks to the Author):

This manuscript describes the history, evolution and domestication of peach through analyzing the deep genome resequencing data of a wide range of cultivated and wild peach species as well as other related species such as cultivated almond. The authors provided genetic and molecular evidence to support that fruit edibility of peach had emerged long before the artificial selection process, mainly contributed by frugivore-mediated selection, and then further improved during the domestication process. The manuscript provided some novel findings regarding peach evolution as well as useful genomic resources for peach breeding and research.

Major concerns

1. I found the first several pages of the manuscript are very hard to understand due to the lack of sufficient background information. Examples are given below:

- a) Lines 26 and 28: mentioning *P. kansuensis* and *P. mira* without providing any background information could cause difficulty in understanding for most of the readers who don't have much peach taxonomy knowledge.
- b) Line 58: it's very hard to follow "Amygdalus species" here without any background information provided.
- c) Line 76-78: Need to provide background information on the selected species and why they were selected, as well as a summary on the number of accessions in each species and which species have edible fruits. I am curious why the author included cultivated almond, *P. pedunculata* and *P. triloba* in the analysis.

Response:

Thanks for the comments and helpful suggestions. To address this issue, we have added some background information about peach taxonomy to help set the stage for the results and ideas described in the manuscript.

For answering questions, **a)** and **b)**, we put the description in Line 53–57:

"*Prunus* subgenus *Amygdalus* comprises two sections, *Amygdalus* and *Persica*, in which domesticated almond and peach are classified, respectively. The common ancestor of this subgenus likely bears a dry, splitting mesocarp, while the transition to fleshy and palatable mesocarp in some *Persica* section species (wild *P. mira* and *P. kansuensis*, and *P. persica*) (Supplementary Table 1) was presumably derived from selection and domestication in China^{8,9},"

For answering the question c), we put the description in Line 78–75:

“Previous studies support that peach was domesticated in China⁴, while the emerged fleshy mesocarp phenotypes are found not only in *P. persica* but also in *P. mira* and *P. kansusensis*. These facts suggested a way for us to explore the selection and/or domestication history that has driven the transition to fleshy edible fruit in some *Persica* section species. In order to clarify the taxonomy of the subg. *Amygdalus* species endemic to China and to study the evolved fruit edibility in peach species, we generated a total of 700 Gb of raw sequence data for 44 high-coverage genomes of cultivated peach and all of the known wild relative species (belonging to *Persica* section) endemic to China, and its closely related species (Supplementary Table 2).”

A summary of the number of accessions in each species and their morphological characters for each species (*e.g.*, edible or inedible fruits, skin color, *etc.*) is provided in Supplementary Tables 1 and Table 4. As noted in our revised text quoted above, our panel included cultivated peach and all of the wild relative species (belong to *Persica* section) endemic to China, as well as cultivated almond (belongs to *Amygdalus* section) and three other closely related relatives (Supplementary Table 2). All of these closely related relative species may help us to understand the evolution of peach. More specifically, *P. mongolica* and *P. tangutica* have been recognized as *Amygdalus* section species; they are closely related to almond (*P. dulcis*). As you may be aware, Charles Darwin actually proposed that peaches are almonds that have been modified in a marvelous manner¹. Our results provide strong molecular evidence against that theory and clarify several other important issues about the evolutionary process of peach species. In addition to the fact that they are prevalent in China, *P. pedunculata* and *P. trilob* have been classified into to a sister clade of the peach clade in a phylogenetic analysis based on morphological data², so we also included these two species in current study and mentioned these issues in the revised manuscript.

References

Darwin, C.R. The variation of animals and plants under domestication, 2nd edn. John Murray, New York, pp 357–441 (1868).

Yazbek, M. & Oh, S.H. Peaches and almonds: phylogeny of *Prunus* subg. *Amygdalus* (Rosaceae) based on DNA sequences and morphology. *Plant Syst. Evol.* **299**, 1403–1418 (2013).

2. One major conclusion, as stated in the abstract, is “fruit edibility was acquired by *P. mira* from an ancient introgression from this common ancestor”. Beside the Treemix analysis, I would suggest addition analysis to obtain more evidence to support the direction of the introgression. The D-statistic the authors used can only detect introgressions, not the direction. The authors should look at partitioned D-statistic described in Eaton and Ree (2013). Inferring phylogeny and

introgression using genomic RADseq data: An example from flowering plants (*Pedicularis*: Orobanchaceae). *Systematic Biology* 62: 689), which does allow for the inference of direction of introgression.

Response:

Thanks for this suggestion about the introgression analysis in our manuscript. Our study of the introgression among peach species was conducted by using both the *D*-statistic and TreeMix model in our manuscript.

Following the suggestion, we have tried to use partitioned *D*-statistic test described in Eaton and Ree (2013. Inferring phylogeny and introgression using genomic RADseq data: An example from flowering plants (*Pedicularis*: Orobanchaceae). *Systematic Biology* 62: 689) to infer the gene flow. However, we found that although it is a good way to infer gene flow direction, the proposed approach does not work technically when applied to our current study due to the unfit phylogenetic relationship. This analysis need a five-group phylogenetic relationship (showed as below): O is the out group, P1 and P2, and P3 and P4, are divergent from their common ancestor P12 and P34 respectively. Like the event showed in this figure, a gene flow from P3 to P2 occurred. Thus we can detect excess alleles (derived alleles shared by P3 and P4) in P2, but not in a gene flow from P2 to P3.

This method can also infer the gene flow from their common ancestor (P12, P34). However, in our case, *P. mira* do not have a sister population. Thus, the recommended method is not applicable under this circumstances.

We then adopted $\phi_a\phi_i$ to estimate the expected gene flow between *P. mira* and the common ancestor of *P. kansuensis* and *P. persica*. The result showed that bidirectional gene flows were found between *P. mira* and the common ancestor of *P. kansuensis* and *P. persica*. However, the gene flow from this common ancestor to *P. mira* was significant stronger than the gene flow from *P. mira* to this common ancestor, supporting the result that we have proposed in the manuscript.

We could also notice that the divergence time of the involved species given by $\phi_a\phi_i$ was inconsistent with the result given by using BEAST approach. Considering that the divergence time from the BEAST approach meets well with the evidence of peach fossils 2.6 mya, is compatible with a series of topographical changes, and is consistent with the previously inferred divergence time of Peach and almond, we decided to emphasize this conclusion in the current study.

3. I am concerned about the choose of wild species used in the selective sweep analysis. The authors used all wild peaches (WP, line 252) including *P. mira*, *P. kansuensis*, *P. mongolica*, *P. tangutica*, and *P. davidiana* to identify selective signatures in the peach genome. This would complicate the identified signatures as some of them are not related to domestication but rather population differentiation, since some of the wild species are not involved in the domestication process at all. I understand that the direct wild progenitor of cultivated peaches remains unclear, but some species which are believed not to be the progenitor (e.g., *P. mira* based on this study?) should be excluded in the analysis to reduce the noises in the results.

Response:

Indeed, since the direct wild progenitor of cultivated peaches remains unclear, the extant peach wild relative species are unique resources available for studying the selection and domestication of peach. This is similar to a case study in soybean (*Glycine Max*) in which its wild relative species *Glycine soja* was used to study the impact of human selection on genetic variation in the soybean genome.

Here, according to this kind suggestion, we added new result by excluding *P. mira* from the selective sweep analysis (using both the SNP and CNV data from current study). We found that the *CNR* genes are remained in the sweep regions derived from these comparisons, supporting our former result that those genes likely play roles in increasing the fruit size during peach domestication. We also identified candidate genes that related to fruit texture, taste, etc., and update our result in Supplementary Table 13–24. It has been reported that the fleshy mesocarp was the result of selection or domestication from the ancestor for the common ancestor of subg. *Amygdalus*¹, in which *P. mira*, *P. kansuensis*, and *P. persica* (*Persica* section species) developed fleshy and palatable mesocarps. Thus, we also retained the results based on all the of the wild relative peach

species (*Persica* section in subg. *Amygdalus*) available in China for selective sweep analysis to avoid the loss of genomic background that existed in peach species.

Reference

1. Yazbek, M. & Oh, S. H. Peaches and almonds: phylogeny of *Prunus* subg. *Amygdalus* (Rosaceae) based on DNA sequences and morphology. *Plant Syst. Evol.* **299**, 1403–1418 (2013).

Minor concerns:

4. “peach (Rosaceae)” in the title is misleading. Rosaceae is a family that peach belongs to.

Response:

We have changed the word “Rosaceae” to “*Prunus persica*” in the title.

5. Line 21: change “Peach is” to “Peach (*Prunus persica* L. Batsch) is”.

Response:

We have change “Peach” to “Peach (*Prunus persica*) is...” in the revised abstract. Now line 19.

6. Line 42, “estimated to 265 Mb”: should be “estimated haploid genome size of 265 Mb”.

Response:

We have change “estimated to 265 Mb” to “estimated haploid genome size of 265 Mb”. Now line 36.

7. Line 52-57: I couldn't follow the logic here. How “indistinguishable endocarp” can suggest “edible fruits” (mesocarp)?

Response:

Thanks for this comment, we now appreciate that the sentence indeed had an ambiguity problem, and we have changed the expression in revised manuscript (Line 46–49), as:

“Surprisingly, peach endocarp fossils from 2.6-million-year-ago (Mya), found recently in Kunming, southwest China, are indistinguishable from endocarps of the modern peach cultivars, and studying of these fossils suggested that peaches may have acquired their modern-like edible fruits long prior to domestication, perhaps mediated by frugivorous primates⁷.”

8. Line 60-61: I don't think fruit size and skin color are fruit edibility traits.

Response:

Yes, it is more appropriate to describe these as “fruit traits” in a category that includes fruit size, texture, taste, and skin color than to use “fruit edibility”. We have changed the expression consistently in our revised manuscript. Also see the description in revised manuscript (Line 57–58):

“Great morphological variation in fruit traits like fruit size, texture, taste, and skin color among cultivated peach and its wild relatives offers a natural diversity panel that presents an opportunity to explore the speciation and domestication history of peach...”

9. Line 78-84: Supplementary Figure 1 is not needed. It’s largely duplicated with Supplementary Table 2 and 4. Supplementary Table 3 should be removed. It’s not necessary. Just need to cite the genome paper or the link to the genome sequence. It’s very strange that in Supplementary Table 5 the numbers in the “Synonymous” column is almost all 15,138.

Response:

We have removed old Supplementary Figure 1 and old Supplementary Table 3, and have provided a link to the peach genome website in method section, and we have fixed the data filling problem in old Supplementary Table 5 (now Supplementary Table 4).

10. Line 100-103: need to cite the corresponding figure for this result.

Response:

We have added the citation for corresponding figure for this result. Now line 117.

11. Line 130, “among these five peach species”: Is almond (*P. dulcis*) a peach species?

Response:

Sorry for this obvious writing error; almond (*P. dulcis*) is not a peach species but rather belongs to *Amygdalus* section of subg. *Amygdalus*. We have changed the expression in revised manuscript (Line 153–156).

“It has been reported that self-fertilization in peach promotes the maintenance of extended LD¹⁸, so the increased extent of LD in the three peach species appears to have been impacted by the inbreeding in the evolutionary history of *Persica* section species.”

12. Line 158: change to “which is comparable to the size of modern peaches while remarkably larger than the fruit of extant non-edible peach species.”

Response:

We have changed “which is comparable to the size of modern peaches and is remarkably larger than the fruit of extant non-edible peach species” to “which is comparable to the size of modern peaches while remarkably larger than the fruit of extant non-edible peach species.” Now line 185.

13. Line 160-162: How the “0.60 Mya” was derived?

Response:

We have added the explanatory text to this sentence in the revised manuscript (now line 188-189) as follow:

“Based on an estimated time to the most recent common ancestor by estimating the divergence time between PL and PMC using all accessions in these two subgroups, we also inferred that the unknown direct wild ancestor of cultivated *P. persica* originated at least 0.60 Mya in the Pleistocene.”

14. Line 168 and 175: I suggest to mark the two periods (115-130 Kya and 19.0-26.5 Kya) on Figure 2b.

Response:

We have marked these two periods on Figure 2b with gray shadows.

15. Line 193: change “The early” to “Early”

Response:

We have changed “The early” to “Early”. Now line 222.

16. Line 194: change “this that” to “this”.

Response:

We have changed “this that” to “this”. Now line 223.

17. Line 348: change “Reads mapping” to “Read mapping”

Response:

We have changed “Reads mapping” to “Read mapping”. Now line 408.

18. Line 351: Provide a reference for SAMtools

Response:

The reference for SAMtools has been added to the reference list and indicated in the method. Now line 411.

19. Line 355: Change “SNPs annotation” to “SNP annotation”

Response:

We have changed “SNPs annotation” to “SNP annotation”. Now line 415.

20. Line 376: Change “reads ratio” to “ratio of reads”

Response:

We have changed “reads ratio” to “ratio of reads”. Now line 436.

21. Line 402-403: How the “substitution rate of 7.7×10^{-9} per site per generation and a generation time of 7 years” was derived?

Response:

A mutation rate $\mu = 7.7 \times 10^{-9}$ per site per generation was derived according to Xie *et al.* (2016), and the generation time of peach was derived from its full reproductive age of around 7 years according Wang *et al.*, 2012. and we mentioned it in the revised manuscript (now line 478-479).

Reference

1. Xie, Z. *et al.* Mutation rate analysis via parent–progeny sequencing of the perennial peach. I. A low rate in woody perennials and a higher mutagenicity in hybrids. *Proc. R. Soc. B.* **283**, 20161016 (2016).
2. Wang, L.R., Zhu, G.R. & Fang, W.C. Peach Genetic Resources in China. *China Agriculture Press*, (2012).

22. Line 428-429, “we assumed 1 cM was equal to 1 M bp”: this seems to be a wrong assumption. Can the authors provide data or references to support this assumption?

Response:

Thanks for your reminding of our inappropriate description here. It is not an assumption with biological significances. Actually, the ADMIXTOOLS took a default to 1Mb = 1 cM without an available linkage map in current study. Thus, we changed the description as: “The significance was assessed by a block jackknife procedure and a block size of 5 Mb was used.” in LINE 498-490. A reference involving a similar analysis was provided as below:

Reference

1. Fu Q, *et al.* An early modern human from Romania with a recent Neanderthal ancestor. *Nature* **524**, 216 (2015).

Reviewer #2 (Remarks to the Author):

The manuscript of Yu et al. (NCOMMS-18-05001) proposes a model for evolution of cultivated peach and related species from its origin in southwest China. The research combines genetic analysis of resequencing data from an ample collection of accessions including mume, almond, various peach wild relatives and cultivated varieties, with archaeological information, providing evidence for several highly relevant aspects of peach history and evolution, some of the most important being : 1) The establishment of the relationships between the group of *Prunus* species analyzed and the identification of introgression patterns between some of them; 2) the sequence of emergence of the different species, and their demographic story in relationship with the geological and climatic events of the different steps of evolution; 3) the evidence that fruit edibility was selected prior to domestication, possibly involving frugivores; 4) the indication that fruit size was selected earlier after

domestication and selection for fruit color is a much more recent event. Overall the paper provides a well-supported and singular evolutionary process for peach, different from that of other well studied perennial species such as apple and grape, and that may represent by itself a model for other tree or herbaceous crops. In addition, the paper provides useful information on candidate genes involved in the inheritance of key economical characters of interest for geneticists and breeders.

My view is that the manuscript describes excellent original research and is acceptable for publication in Nature Communications. I have only two comments and a few minor suggestions for improvement that are described below.

An interesting suggestion of the manuscript is that monkeys may have played a role in the selection for fruit edibility in peach and *Prunus mira* before domestication. As evidence for this, the Ne trajectories of three species during the Pleistocene are provided, two of which possibly sharing the same habitat with peach and another species that did not. The authors consider that the two former species shared similar trajectories but not the third, which is something that was not evident to me just by looking at Figure 2. A more detailed explanation or a statistical test supporting this trajectory similarity would help to make this stronger evidence.

Response:

Thanks for your enthusiasm and comments generally and for this kind suggestion in particular. We therefore added this result in the revised manuscript (now line 271-275) as:

“Subsequently, we used a Granger causality (GC) test³⁸ to assess if there is a similar trend in the effective population size (N_e) between the three monkey species and the peach species. Intriguingly, our GC results implied that the N_e of the two southwest monkey species (*Rhinopithecus brelichi* and *R. bieti*) can be used to predict the N_e trajectories of the three edible peach species (*P. mira*, *P. kansuensis*, and *P. persica*) (Supplementary Note 3).”

The authors identify a number of putative selection sweeps comparing the wild, landrace and modern cultivar compartments, and then find in these regions genes involved in fruit characteristics of agricultural interest. Some relevant examples are provided suggesting selection for genes known to affect these characters in other species that are located in the selection sweeps. This is a reasonable approach, but one could argue that provided a sufficiently large number of candidate genes, the probability of finding one or several of these genes by chance in the genome fragments identified is high. Would it be possible for one or more of these characters to estimate the total number of genes possibly involved and to test whether the numbers of genes identified are significantly higher than those expected by chance?

Response:

Thanks for your comment. Working from the assumption that the selected fruit-related genes are enriched in the selective sweep regions, we estimated the putative “threshold number” for fruit-related genes in the peach genome: using Fisher's exact test (significant P -value < 0.05), one would expect no more than 1,296 (WP vs. CP), 1,626 (WP vs. PL), or 576 (PL vs. PMC) fruit-related genes should present on peach genome. However, given that there is no designated ‘fruit-related’ gene list, it is hard to estimate the exact significance of the putative selected fruit genes number in sweep regions. Note that we chose candidate genes from among the top 1% of selective sweep regions, so there was a strict and reliable selection cutoff in our study of the domestication of fruit traits. Also, consider that a limited number of key genes may play hugely influential roles in driving a “domestication syndrome” during crop domestication; under this circumstance, we would not expect a significantly higher number of genes in a candidate sweep region.

Minor changes proposed:

Abstract

Line 23: change “... species are closely...” to species “... species is closely...”

Response:

We have changed “... species are closely...” to species “... species is closely...”. Now line 70 in the main text.

Line 31: change “...during peach...” to “...during modern peach...”

Response:

We have changed “...during peach...” to “...during modern peach...”. Now line 76 in the main text.

Introduction

Line 72. "...our novel hypothesis...". This hypothesis was already proposed in the paper of Su et al. (2015), cited in the manuscript.

Response:

We have modified the expression in this sentence and cited the paper. Now line 71-73.

Results

Line 97. Change "...maturity and morphologically..." to "...maturity and are morphologically..."

Response:

We have changed "...maturity and morphologically..." to "...maturity and are morphologically...".
Now line 112.

Line 194. Change "...and this that..." to "...and this..."

Response:

We have changed "...and this that..." to "...and this...". Now line 223.

Line 233. Change "...the basis characteristics..." to "...the basic characteristics ..."

Response:

We have changed "...the basis characteristics..." to "...the basic characteristics ...". Now line 262.

Lines 283-284. A reference could be provided to support the statement that stone and fruit size are strongly correlated.

Response:

The correlation between the size of a peach stone and fruit size has been reported in Zheng *et al.*, 2014, and we added the citation to this conclusion in the revised manuscript. Now line 322.

Methods

Line 421. Change "...based genome-wide..." to "...based on genome-wide..."

Response:

We have changed "...based genome-wide..." to "...based on genome-wide...". Now line 483.

Figure Legends

Line 676. Change "...Linage..." to "...Linkage..."

Response:

We have changed "...Linage..." to "...Linkage...".

Supplementary Figure 6. The *P. ferganensis* group appears to be the one marked with dark green color, but it is not sufficiently clear. It could be stated on the legend or indicated on the diagram.

Response:

We have modified the old Supplementary Figure 6 (now the Supplementary Figure 5) and added the statement to the legend following the reviewer's helpful suggestion.

Reviewer #3 (Remarks to the Author):

The authors used 58 high-coverage genomes of cultivated peach and closely related relatives to explore the evolutionary history, including the genetic divergence of cultivated peaches and its closely related relatives, demographic trajectories, domestication features and their evolved time. The topics addressed are interesting, but there are several problems with the manuscript as presented.

The exploration of data analysis with appropriate tools is quite weak and some of the conclusions are lack of evidences to support. There are several issues with the manuscript that need to be addressed before the manuscript is published.

Major issues:

1. SNP calling and filtering is not clear enough. The author reported that the total SNPs called are 24,280,369, but it is unclear why only use 2,559,589 SNPs to do PCA and 3,909,617 to build NJ tree and how they did filtering. Please also clarify how the author did SNP validation with SNP array data and how many of those SNPs can be found in the resequencing data. In order to see if the SNPs called from whole genome resequencing data are reasonable, please show the site frequency spectrum (SFS) and the ratio of transition and transversion (Ts:Tv). Since the author did annotations for all of the SNPs with Annovar, but it is unclear how many of those SNPs are noncoding, coding, synonymous or nonsynonymous SNPs.

Response:

Thanks for the comment. We generated two sets of results: **1)** firstly using all 58 accessions and *P. mume* (Supplementary Fig. 2), and **2)** then removed the non-*Amygdalus* accessions for the 51 *Amygdalus* accessions (Fig. 1b,c). The total SNPs were called based on all 58 accessions which included the species that showed large genetic differences compared to the reference genome of peach, leading to a total of 24,280,369 SNPs. The PCA and NJ tree in Fig. 1 was built based on the SNPs from 51 *Amygdalus* accessions, which have been filtered as follows: PCA: maf < 5%, call rate > 90%; NJ-tree: maf < 5%, call rate > 90% (also mentioned in method).

We have developed an Axiom® Peach 170K SNP array with SNPs discovered in 96 peach accessions (including the accessions in this study, and from two previous studies, Verde *et al.*, 2013 and Cao *et al.*, 2014). Since this SNP array was developed based on the *P. persica* Whole Genome v1.0 reference genome, we thus applied the liftover program to transit the genotyped SNPs from *P. persica* Whole Genome v1.0 reference genome to *P. persica* Whole Genome v2.1.a1 for SNP validation as we described in the manuscript.

We have added the analysis of SFS (distribution of MAF) (Supplementary Fig. 8) and addressed this result in the revised manuscript. The number of Ts and Tv and the value of Ts:Tv for each accession were calculated and are provided in Supplementary Table 4. All SNP annotation information for each accession is also presented in Supplementary Table 4; this contains

information including noncoding, coding, synonymous or nonsynonymous SNPs, the Ts/Tv ratio, etc.

2. Some of the analyses applied are either potentially inappropriate or over interpreted. For example, the authors make use of ADMIXTURE for population structure. However, ADMIXTURE likely makes a Hardy Weinberg assumption regarding expected genotypes. Violations of this assumption could lead to over clustering of samples owing to inbreeding. Did the authors have a strategy for addressing this in samples that are highly inbred?

Response:

Thanks for this comment. ADMIXTURE assumes that loci are at HWE within populations, and many programs that use allele frequencies to analyze genetic data assume HWE at some point, and it is now appreciated that HWE is seldom met in practical studies. Thus, we indeed need to cautiously interpret our results by combining more additional analysis. We therefore also included PCA, NJ-tree, IS, IBS, and MJ-networking analysis to manage any spurious findings that could have come from any one particular method with the goal of more comprehensively addressing the population structure of these species.

In order to reduce this affect, we filtered SNPs by testing the HWE violation ($P > 10^{-4}$) and reconstructed the model-based clustering result as follow:

This model-based clustering result ($K = 9$) also supported the grouping pattern that given by NJ tree, and we could also draw the same conclusion that *P. ferganensis* is a geographical ecotype of *P. persica*, as the *P. ferganensis* accessions were clustered with the *P. persica* landraces from north China ($K = 7$).

In summary, we believed that the ADMIXTURE tool, as has been widely used in population structure study of many domesticated crops including species with extreme high inbreeding levels like rice (a heterozygosity of 0.0038%)¹, does provide valuable information to investigate the population structure. However, we agree and have emphasized in the writing of our revised manuscript that cautious interpretation should be addressed by combining additional analyses and more biological background.

Reference

1. Du, H. *et al.* Sequencing and de novo assembly of a near complete indica rice genome. *Nat. Commun.* **8**, 15324 (2017).

3. The authors used VCFtools to calculate genetic nucleotide diversity, Tajima's D and Fst. VCFtools were designed to calculate the population genetic statistics for diploid samples, but the majority of the samples in this study are highly selfing (which will be treated as haploid), and cannot use VCFtools to calculate genetic nucleotide diversity, Tajima's D and Fst.

Response:

Kindly note that the other major model species for fruit genomics research—apple and grape—are self-incompatible, so we were guided in our analysis by studies in the self-compatible model plant rice. Although *P. persica* is self-compatible, it shows a higher heterozygosity (0.126%) than human (a heterozygosity of 0.43×10^{-3})^{1,2}. While, in studies of human, samples are not typically treated as haploid for the calculation of nucleotide diversity, Tajima's D, or *Fst* values.

References:

1. Wang, J. *et al.* The diploid genome sequence of an Asian individual. *Nature* **456**, 60–65 (2008).

2. Li, R. *et al.* SNP detection for massively parallel whole-genome resequencing. *Genome Res.* **19**, 1124–1132 (2009).

4. All of the scripts and analytical tools used for a largely computational project like the work reported here need to be made available through Github, Bitbucket, or a similar version control repository. Because the scripts are essential to reproducing this work, they should be available to reviewers and/or readers. These resources are important in part because samtools, GATK, and similar utilities include a large number of options that alter the resulting output.

Response:

According to your kind advice, we have uploaded the primary code used in the current study to Github:

https://github.com/ColyFu/peach_edibility/blob/master/work.sh

5. Some of the arguments or conclusions seem to be lack of the evidences to support. For example,

in Line 129, the authors argued that "it appears that inbreeding level has likely played a major role in determining the level of LD among these five peach species", but apparently the authors did not examine the influences of other factors except inbreeding. Another example is in line 237 to line 245, the authors proposed that " the ancient selection process likely involved frugivorous, contributing to the emergence of fruit edibility, prior to recent human-mediated domestication" only based on that the Ne trajectories of three monkey species has the similar pattern with the peaches in this study.

Response:

Thanks for your comments. Yes, Linkage disequilibrium is influenced by many factors, including selection, rate of recombination, mutation, genetic drift, mating system, population structure, and genetic linkage, and it has been reported that mating system differences between closely related species pairs can significantly affect many aspects of genome evolution in *Arabidopsis*, *Capsella*, and *Collinsia* (including lower nucleotide diversity, higher linkage disequilibrium (LD), and reduced effective population size (N_e))¹. For *Prunus specis*, it is believed that SI is predominant although self-compatibility (SC) forms occur in *P. persica*. According to our results, the increasing in the inbreeding level in *P. mira*, *P. kansuensis*, and *P. persica* (as evidenced by ROH lengths and F values) likely contribute to the high extent of LD in these species. This conclusion also mirrors a previous study² which proposed that self-fertilization in peach favors the maintenance of extended LD. It is of course complicated to fully interpret the evolutionary consequences of the relationship between selection/domestication, inbreeding, and LD, thus, we have change the expression in the revised manuscript (now line 134-156).

References

1. Velasco, D. *et al.* Evolutionary genomics of peach and almond domestication. *G3: Genes, Genomes, Genetics* **6**, 3985-3993 (2016).
2. Cao, K. *et al.* Genome-wide association study of 12 agronomic traits in peach. *Nat. Commun.* **7**, 13246 (2016).

Thanks for your comments. We thus performed Granger causality test for assessing N_e trend similarity between monkey and peach species to provide more statistical support for our proposal that the pre-historical animal-plant interaction for peach is plausible in the revised manuscript (now line 271-275) as:

“Subsequently, we used a Granger causality (GC) test³⁸ to assess if there is a similar trend in the effective population size (N_e) between the three monkey species and the peach species. Intriguingly, our GC results implied that the N_e of the two southwest monkey species (*Rhinopithecus brelichi* and *R. bieti*) can be used to predict the N_e trajectories of the three edible peach species (*P. mira*, *P. kansuensis*, and *P. persica*) (Supplementary Note 3).”

Previous studies have proposed that the ancestry of the common ancestor of *Amygdalus* subgenus species had dry, split, inedible fruit, and was most self-incompatible, while the transition to fleshy edible mesocarp seems to have resulted from the selection/domestication traits in peach species^{1,2}. We also found “genomic footprints” that likely result from positive selection in the three edible peach species. We added some new context in revised manuscript in line 126-129 and line 143-149.

Additionally, the study of peach fossils evidence implies that frugivores likely selected peaches prior to a later human mediated domestication³. As a matter of fact, it is commonly found in zooecology studies that frugivores (also known as dispersers) can lead dispersal syndromes for fruit traits^{4,5}. We thus discuss the idea that frugivores may have influenced the aforementioned pre-history selection by contributing to the emergence of fleshy fruit in peach species in the revised manuscript, and we also added some viewpoints from previous studies for a fuller discussion of the evolution of fleshy mesocarp in peach species, hopefully thereby adding valuable information about this issue. See more additional details in discussion section in revised manuscript.

Reference

1. Yazbek, M. & Oh, S. H. Peaches and almonds: phylogeny of *Prunus* subgenus *Amygdalus* (Rosaceae) based on DNA sequences and morphology. *Plant Syst. Evol.* **299**, 1403-1418 (2013).
2. Yazbek, M. & Al-Zein, M. S. Wild almonds gone wild: revisiting Darwin’s statement on the origin of peaches. *Genet Resour Crop Evol.*, **61**, 1319-1328 (2014).
3. Su, T., Wilf, P., Huang, Y., Zhang, S. & Zhou, Z. Peaches preceded humans: fossil evidence from SW China. *Sci. Rep.* **5**, srep16794 (2015).
4. Lomáscolo, S. B. *et al.* Correlated evolution of fig size and color supports the dispersal syndromes hypothesis. *Oecologia* **156**, 783-796 (2008).
5. Valenta, K. *et al.* Colour and odour drive fruit selection and seed dispersal by mouse lemurs. *Sci. Rep.* **3** 2424 (2013).

6. The author argued that " it is reasonable to infer that common ancestor of *P. persica* and *P. kansuensis* was self-incompatible. We therefore propose that the combination-incompatibility and overlapping habits...approximately 3 million years ago". Is *P. kansuensis* self-incompatible? Why is it reasonable to infer that common ancestor of *P. persica* and *P. kansuensis* was self-incompatible? Again it is hard to follow the statement that "We therefore propose that the combination-incompatibility and overlapping habits...approximately 3 million years ago". How did the author get this conclusion?

Response

Thanks for your comments, we have modified our description in the new manuscript (line 242-249) as follow:

“*P. persica* has a high level of inbreeding owing to its self-compatibility. It has been proposed that self-incompatibility (SI) is predominant in *Prunus*⁹, and considering that the transition to self-compatibility (SC) is known to have been caused by a loss-of-function mutation³², we infer that populations of the common ancestor of *P. kansuensis* and *P. persica* likely contained some SI individuals that would have enabled cross-pollination. We therefore propose that the combination of a capacity for cross-pollination and overlapping habitats (before the outwards dispersal and domestication of *P. persica*) created a highly favorable context for the ancient introgressions between *P. mira* and the common ancestor of *P. persica* and *P. kansuensis* approximately 3 million years ago.”

7. Line 221-223, how did the author know stronger signatures in introgressed segments compared with genome levels of *P. mira*? Based on which comparison?

Response:

We have added the description to the revised manuscript. The comparison was performed using a bi-population selective sweep method, XP-EHH, and the *P*-value is presented in the main text (line 250-252).

Minor issues:

1. Line 47, please fix the citation format.

Response:

We modified this citation format and added the reference (now line 41).

2. Line 114, "An increase in the ratio of nonsynonymous to synonymous SNPs." in line 114, did the author use the private SNPs to do calculation?

Response:

No. We used all of the SNPs for the analysis of the ratio of nonsynonymous to synonymous SNPs.

3. Which tools did the author to calculate the ROH? Please clarify in the Method.

Response:

We used PLINK for the calculation, and we have provided the information about the calculation of ROHs in the methods section (line 443-449).

4. Which tool did the authors calculate the inbreeding coefficient? Please clarify in the Method.

Response:

The coefficient of inbreeding (*F* value) in each population was calculated using the formula $F = 1 - H_o/H_e$, and we have clarified the calculation of inbreeding coefficient in method (line 450-452).

5. Please cite the paper for the substitution rate of 7.7×10^{-9} in the main text.

Response:

We have added the description of the substitution rate (7.7×10^{-9}) in the main text (now line 159).

6. Please label the samples consistently. For example, use those labels CA, WM, WD, WK, CP in all of the figures, tables, and supplemental tables, otherwise it is hard to follow.

Response:

We re-labeled the samples consistently by changing their captions appropriately with WM, WD, or WK, while for CP (cultivated peach, *P. persica*) and CA (cultivated almond, *P. dulcis*), both of which included modern cultivar and landrace accessions, we refer to them as: PL (*Persica* landrace) and PMC (*Persica* modern cultivar), and DL (*Dulcis* landrace), and DMC (*Dulcis* modern cultivar) in the revised manuscript.

Response to Reviewers' comments

The second review

Reviewer #1 (Remarks to the Author):

The authors addressed all my concerns. Here are some minor changes I would suggest:

Line 19: change “for study” to “for studying”

Response:

Thanks for your continued input about our study; we appreciate your help and guidance. We have changed “for study” to “for studying” in the revised abstract. Now line 19.

Line 20: Change “We here explored” to “Here we explore”

Response:

We have changed “We here explored” to “Here we explore” in the revised abstract. Now line 20.

Line 29: “an indispensable instance” seems not accurate. May change to “valuable information”.

Response:

We have changed “an indispensable instance” to “valuable information” in the revised abstract. Now line 28-29.

Line 41: changed “with annual” to “with an annual”

Response:

We have changed “with annual” to “with an annual” in the revised manuscript. Now line 40-41.

Line 68: Reference 9 seems wrongly cited here. Please check the reference citation across the manuscript.

Response:

Thanks for your bringing this to our attention. We have re-checked reference 9: “Wild almonds gone wild: revisiting Darwin’s statement on the origin of peaches”, in which the author wrote that “Alternatively, the fleshy mesocarp characterizing the peach subclade may be a byproduct of domestication. Many studies suggest that *P. mira*, *P. ferganensis* and *P. kansuensis* are feral types of the cultivated peach, supporting the hypothesis that the loss of a dry splitting mesocarp is the result of artificial selection for fleshiness during domestication”.

Line 90: remove “with GATK”.

Response:

We have removed “with GATK” in the revised manuscript.

Line 93: change “principle” to “principal”

Response:

We have changed “principle” to “principal” in the revised manuscript. Now line 92.

Line 133: Remove “data”

Response:

We have removed “data” in the revised manuscript.

Line 181: Change “accurate” to “the case”

Response:

We have changed “accurate” to “the case” in the revised manuscript. Now line 183.

Line 196: Change “began” to “starting”

Response:

We have changed “began” to “starting” in the revised manuscript. Now line 198.

Line 223 and 252: add references for TreeMix and XP-EHH

Response:

We added the references for TreeMix and XP-EHH in the revised manuscript. Now line 224 and 246, respectively.

Line 262: Change “basis” to “basic”

Response:

We have changed “basis” to “basic” in the revised manuscript. Now line 256.

Line 288: remove “and this especially so in peach”

Response:

We have removed “and this especially in peach” in the revised manuscript. Now line 281.

Thanks again for your work on our behalf.

Reviewer #2 (Remarks to the Author):

The authors have addressed correctly my comments from the first review. I think that the current versión is aceptable for publication in Nature Communications.

I have found a few minor errors that should be corrected by the authors:

Line 93 and legend of Supp Fig 2: change principle to principal

Response:

We appreciate your ongoing support and are thankful for your continued guidance for our study.

We have changed “principle” to “principal” in the revised manuscript. Now line 92 and legend of Supplementary Fig. 2.

Line 138 delete "Rosaceae"; grapewine is nor a rosaceous crop

Response:

Sorry for this obvious error. We have deleted "Rosaceae" in the revised manuscript. Now line 142.

Line 384 change "process" to "processes"

Response:

We have changed "process" to "processes" in the revised manuscript. Now line 376.

Thanks for your continued help in improving our study.

Reviewer #3 (Remarks to the Author):

The author solved majority of the my comments, but some of the explanation are not reflected in the main text. Also a couple of my comments are not well explained.

Unsolved major issues:

1. SNP calling and filtering is not clear enough. The author reported that the total SNPs called are 24,280,369, but it is unclear why only use 2,559,589 SNPs to do PCA and 3,909,617 to build NJ tree and how they did filtering. Please also clarify how the author did SNP validation with SNP array data and how many of those SNPs can be found in the resequencing data. In order to see if the SNPs called from whole genome resequencing data are reasonable, please show the site frequency spectrum (SFS) and the ratio of transition and transversion (Ts:Tv). Since the author did annotations for all of the SNPs with Annovar, but it is unclear how many of those SNPs are noncoding, coding, synonymous or nonsynonymous SNPs.

Original Response: "Thanks for the comment. We generated two sets of results: 1) firstly using all 58 accessions and P. mume (Supplementary Fig. 2), and 2) then removed the non-Amygdalus accessions for the 51 Amygdalus accessions (Fig. 1b,c)."

Please reflect this in the caption of Fig. 1b

Response:

Thanks for your continued guidance to improve our manuscript. As for this suggestion. We now present these details in the caption for Fig. 1b.

Original Response: "The total SNPs were called based on all 58 accessions which included the species that showed large genetic differences compared to the reference genome of peach, leading to a total of 24,280,369 SNPs. The PCA and NJ tree in Fig. 1 was built based on the SNPs from 51 Amygdalus accessions, which have been filtered as follows: PCA: maf < 5%, call rate > 90%; NJ-tree: maf < 5%, call rate > 90% (also mentioned in method)."

Please clarify the filtering criteria in the caption of Fig. 1b.

Response:

Thanks for this suggestion. We have clarified the filtering criteria in the caption of Fig. 1b.

Original Response: "We have developed an Axiom® Peach 170K SNP array with SNPs discovered in 96 peach accessions (including the accessions in this study, and from two previous studies, Verde et al., 2013 and Cao et al., 2014). Since this SNP array was developed based on the P. persica Whole Genome v1.0 reference genome, we thus applied the liftover program to transit the genotyped SNPs from P. persica Whole Genome v1.0 reference genome to P. persica Whole Genome v2.1.a1 for SNP validation as we described in the manuscript."

Please add this in the method part of the main text or at it in the supplemental notes.

Response:

We have added this SNP validation details in the Methods section of the main text. Now line 406-412.

Original Response: “We have added the analysis of SFS (distribution of MAF) (Supplementary Fig. 8) and addressed this result in the revised manuscript. The number of Ts and Tv and the value of Ts:Tv for each accession were calculated and are provided in Supplementary Table 4. All SNP annotation information for each accession is also presented in Supplementary Table 4; this contains information including noncoding, coding, synonymous or nonsynonymous SNPs, the Ts/Tv ratio, etc.”

2. Some of the analyses applied are either potentially inappropriate or over interpreted. For example, the authors make use of ADMIXTURE for population structure. However, ADMIXTURE likely makes a Hardy Weinberg assumption regarding expected genotypes. Violations of this assumption could lead to over clustering of samples owing to inbreeding. Did the authors have a strategy for addressing this in samples that are highly inbred?

Original Response: “Thanks for this comment. ADMIXTURE assumes that loci are at HWE within populations, and many programs that use allele frequencies to analyze genetic data assume HWE at some point, and it is now appreciated that HWE is seldom met in practical studies. Thus, we indeed need to cautiously interpret our results by combining more additional analysis. We therefore also included PCA, NJ-tree, IS, IBS, and MJ-networking analysis to manage any spurious findings that could have come from any one particular method with the goal of more comprehensively addressing the population structure of these species.”

Original Response: “In order to reduce this affect, we filtered SNPs by testing the HWE violation ($P > 10^{-4}$) and reconstructed the model-based clustering result as follow:...”

Please add this part in the Method of the main text and update the Supplementary fig5

Response:

We have added the model-based clustering analysis that used filtered SNPs and tested HWE violations in the Methods section of the main text (Now line 425-427); we have also updated Supplementary Fig. 5 according to your helpful suggestion.

*Original Response: “This model-based clustering result ($K = 9$) also supported the grouping pattern that given by NJ tree, and we could also draw the same conclusion that *P. ferganensis* is a geographical ecotype of *P. persica*, as the *P. ferganensis* accessions were clustered with the *P. persica* landraces from north China ($K = 7$).”*

Original Response: “In summary, we believed that the ADMIXTURE tool, as has been widely used in population structure study of many domesticated crops including species with extreme high inbreeding levels like rice (a heterozygosity of 0.0038%)1, does provide valuable information to investigate the population structure. however, we agree and have emphasized in the writing of our revised manuscript that cautious interpretation should be addressed by combining additional analyses and more biological background.”

I did not see you emphasized in the revised version. Please reflect in the main text and point it out in the Rebuttal letter.

Response:

Thanks for your helpful suggestion. Considering the potential HWE violation, we have added the result of model-based clustering analysis using filtered SNPs by testing the HWE violation and have updated the main text of the paper to more comprehensively address this important concern (Now line 120-125). The updated text for this point from the main manuscript reads as follows:

“ADMIXTURE, which is widely used and provides valuable information for investigating the population structures of many domesticated crops—including species with extreme high inbreeding level like rice¹⁶—assumes that loci are at HWE (Hardy-Weinberg equilibrium) within populations. Considering the potential impact of HWE violations, we further filtered SNPs by testing HWE violation ($P > 10^{-4}$) and performed model-based clustering analysis; this analysis gave to a consistent conclusion about the population structure ($K = 9$; Supplementary Fig. 5b).”

6. The author argued that " it is reasonable to infer that common ancestor of *P. persica* and *P. kansuensis* was self-incompatible. We therefore propose that the combination-incompatibility and overlapping habits...approximately 3 million years ago". Is *P.kansuensis* self-incompatible? Why is it reasonable to infer that common ancestor of *P.persica* and *P.kansuensis* was self-incompatible? Again it is hard to follow the statement that "We therefore propose that the combination-incompatibility and overlapping habits...approximately 3 million years ago". How did the author get this conclusion?

Original Response: "Thanks for your comments, we have modified our description in the new manuscript (line 242-249) as follow:

"P. persica has a high level of inbreeding owing to its self-compatibility. It has been proposed that self-incompatibility (SI) is predominant in Prunus⁹, and considering that the transition to self-compatibility (SC) is known to have been caused by a loss-of-function mutation³², we infer that populations of the common ancestor of P. kansuensis and P. persica likely contained some SI individuals that would have enabled cross-pollination. We therefore propose that the combination of a capacity for cross-pollination and overlapping habitats (before the outwards dispersal and domestication of P. persica) created a highly favorable context for the ancient introgressions between P. mira and the common ancestor of P. persica and P. kansuensis approximately 3 million years ago."

The author did not answer my question " Is *P.kansuensis* self-incompatible?". Also does the SI locus in the introgression region? What does "overlapping habitats" mean? Please clarify.

Response:

Thank you for focusing our attention on this aspect of our manuscript. In the course of our revision process, we eventually came to the conclusion that the best course of action was to remove these speculative claims from our manuscript; we have done this. Here, seeking to address

your questions about our initially included speculative claim, we simply revisit the data which had motivated our now-removed conclusions, and present the answers as follow:

The author did not answer my question " Is *P.kansuensis* self-incompatible?"

Response:

Specifically, whereas *P. kansuensis* is regard as a self-compatible species^{1,2}, we found that both *P. mira* and *P. kansuensis* showed lower inbreeding level than *P. persica*, but higher inbreeding level than *P. dividiana* and *P. dulcis* (Fig. 1d).

We initially interpreted this to indicate that perhaps relatively more cross-pollination had occurred in these two peach species. We then surmised that, given both the proposal that self-incompatibility (SI) is predominant in *Prunus* and the evidence of a gradual transition to self-compatibility (SC) in some *Prunus* section peach species, that the common ancestor of *P. kansuensis* and *P. persica* could have had a lower inbreeding level than *P. persica*. This would have, in theory, enabled more cross-pollination and would have potentially encouraged the introgression between *P. mira* and the common ancestor of *P. kansuensis* and *P. persica*.

References:

1. Zheng, Y., Crawford, G. & Chen, X. Archaeological evidence for peach (*Prunus persica*) cultivation and domestication in China. *PLoS One*, e106595 (2014).
2. Velasco, D., Hough, J., Aradhya, M. & Rossibarra, J. Evolutionary genomics of peach and almond domestication. *G3: Genes/Genomes/Genetics*. 6, 3985-3993 (2016).

Also does the SI locus in the introgression region?

Response:

We scanned the introgressed segments, and found that the S-locus is not involved in this introgression.

What does "overlapping habitats" mean?

Response:

The term that we had used but have now removed from our manuscript ("overlapping habitats") was intended to refer to a coinciding area in which both the *P. mira* and *P. kansuensis* habitats were located (northeast of the Tibetan plateau, south of current Gansu province, and west of current Sichuan province) (Fig.1a). Hence, we had proposed that "the combination of a capacity for cross-pollination and overlapping habitats (before the outwards dispersal and domestication of *P. persica*) created a highly favorable context for the ancient introgressions between *P. mira* and the common ancestor of *P. persica* and *P. kansuensis*".

Guided by your concerns, we now appreciate that this speculative conclusion was not strong enough to merit inclusion in our manuscript. Clearly, further studies will be needed to more comprehensively address SI or SC in peach species.

We again thank the reviewer for their guidance and care in helping us to improve our manuscript.

Response to Reviewers' comments

The third review

Reviewer #3 (Remarks to the Author):

The authors have addressed correctly my comments from the second review. I think that the current version is acceptable for publication in Nature Communications.

Response:

We deeply appreciate your ongoing support, and are thankful for your guidance throughout the improvement process for our study.